# Joint Resource Slicing and Vehicle Association for Drone-Assisted Vehicular Networks †

**Hang Shen \*** , **Tianjing Wang** , **Yilong Heng and Guangwei Bai**

College of Computer and Information Engineering (College of Artificial Intelligence), Nanjing Tech University, Nanjing 211816, China

\* Correspondence: hshen@njtech.edu.cn

† Part of this work has been published in IEEE Symposium on Computers and Communications (ISCC).

**Abstract:** The drone-small-cell-assisted air-ground integrated network is a promising architecture for enabling diverse vehicle applications. This paper presents a joint resource slicing and vehicle association framework for drone-assisted vehicular networks, which facilitates spectrum sharing among heterogeneous base stations (BSs) and achieves dynamic resource provisioning in the presence of network load dynamics. We formulate the network utility maximization problem as mixed-integer nonlinear programming, considering traffic statistics, quality-of-service (QoS) constraints, varying vehicle locations, load conditions in each cell, and interdrone interference. The original maximization problem is transformed into a biconcave optimization problem to ensure mathematical tractability. An alternate concave search algorithm is then designed to iteratively solve vehicle association patterns and spectrum partitioning among heterogeneous BSs until convergence. Simulation results show that the proposed scheme achieves a significant performance improvement in throughput and spectrum utilization compared with two other baseline schemes.

**Keywords:** vehicular networks; drone; spectrum slicing; resource allocation; quality-of-service (QoS)





## 1. Introduction

As a typical fifth-generation (5G) and beyond scenario, vehicular networks connect vehicles, pedestrians, mobile devices, and base stations (BSs), providing a wide range of services, such as road safety and security, fleet, and traffic management [1]. Cellular vehicle-to-everything (C-V2X) is seen as a viable enabler for emerging use cases in 5G and beyond, offering low-latency, high-reliability, and high-throughput communications for various services and supporting massively interconnected vehicles [2]. However, increasing network capacity and accommodating various services with stringent quality-of-service (QoS) requirements necessitate innovations in network architecture. Ultradense deployment of BSs and roadside units can improve network capacity through network densification. Still, this solution may lead to low cell utilization efficiency, additional infrastructure deployment costs, and intercell interference [3]. Providing satisfactory QoS with fixed and rigid terrestrial cellular networks, particularly on busy urban roads during rush hours, is challenging.

Drones equipped with specialized wireless transceivers and computing modules have the potential to form drone-based small cells (DSCs). These flying base stations (BSs) can communicate with ground-based base stations (GBSs) and provide vehicle connectivity services. An integrated air-ground network architecture is promising for enabling ubiquitous connections, enhancing the performance of 5G and beyond vehicular networks. First, DSCs can fly in designated areas to form three-dimensional, configurable small cells. These small cells can be rapidly deployed to cover wireless "dead zones" for vehicular users, enabling better network extensibility. Second, combining GBSs with DSCs can provide broader coverage and greater capacity through a three-dimensional layered network. DSCs

can act as airborne relays for GBSs to serve edge vehicles beyond GBS coverage. Third, drones can monitor traffic conditions in the air (including road congestion and accidents) and transmit this information to ground stations to be relayed to vehicular networks, which helps drivers choose optimal routes. Last but not least, DSCs flying at high altitudes avoid shadow fading in data transmission and increase the probability of establishing reliable short-distance line-of-sight (LoS) links, reducing delays and improving reliability [4]. Deploying DSCs under GBSs facilitates spectrum reuse owing to their low transmit power and flexible placement, relieving resource allocation pressures.

Resource allocation issues arise despite the numerous benefits of an integrated air-ground network architecture. First, the unique GBS-to-DSC (G2D) and DSC-to-vehicle (D2V) channels create a trade-off between effective coverage and spectrum utilization in DSC deployment. Resource allocation and vehicle association must consider DSC's effective coverage impact. Second, the maneuverability of DSCs results in diverse vehicle association patterns, complicating spectrum slicing among heterogeneous base stations. Third, interference fluctuations in DSC deployment [5] make granular vehicle-level resource provisioning difficult. The movement of DSCs creates rapidly changing interference conditions and network topology. Fourth, DSC deployment and position adjustment should account for road direction and traffic variation. For example, DSCs should fly along the direction of high-speed vehicular traffic for better service coverage. Therefore, exploring efficient spectrum resource provisioning that cooperates with DSCs to support emerging vehicular applications is crucial.

Resource allocation issues arise despite the numerous benefits of an integrated air-ground network architecture. First, the unique ground-base-station-to-drone-small-cell (G2D) and drone-small-cell-to-vehicle (D2V) channels create a trade-off between effective coverage and spectrum utilization in DSC deployment. Resource allocation and vehicle association must consider DSC's effective coverage. Second, the maneuverability of DSCs results in diverse vehicle association patterns, complicating spectrum slicing among heterogeneous base stations. Third, interference fluctuations in DSC deployment [5] make granular vehicle-level resource provisioning difficult. The movement of DSCs creates rapidly changing interference conditions and network topology. Fourth, DSC deployment should account for road direction and traffic variation for better service coverage.

### 1.1. Related Works

Many resource slicing methods are designed for terrestrial vehicular networks, highlighting service provision capability and QoS satisfaction for various services. Peng et al. in [6] developed a joint power control and resource slicing strategy to provide QoS-guaranteed downlink transmissions in multiaccess edge computing (MEC)–enabled vehicular networks. They also proposed a multidimensional resource management framework in [7] to maximize the number of offloaded tasks under heterogeneous QoS requirements. In [8], a multitimescale radio access network slicing and task offloading problem is investigated to maximize resource utilization with diverse QoS guarantees for autonomous driving tasks. Flexible wireless resource management is explored in [9], where radio access and processing functions run in software instances based on network function virtualization (NFV) [10]. Shen et al. in [11] proposed a network architecture that facilitates the interplay between the digital twin and network slicing paradigms, building on holistic network virtualization and edge intelligence. Zarandi and Tabassum in [12] investigated the delay minimization problem with task offloading, computation, and communication resource allocation in sliced multicell mobile edge computing (MEC) systems. They solved offloading decision-making and resource allocation subproblems through alternating optimization until convergence. A reinforcement learning method is developed in [13] for the decision making of network selection and autonomous driving in multiband vehicular networks, with the goal of enhancing the data rate through radio resource management.

Utilizing DSCs is crucial for effective service provisioning in vehicular networks. Recent research has focused on device association, DSC coverage, and resource allocation. Sun et al. in [14] examined the spectrum efficiency at end devices and explored how DSC deployment can enhance resource utilization. Shi et al. in [15] developed a drone ground coverage model to maximize end device coverage while adhering to the drone-to-ground link quality constraint. However, the impact of drone flight height on resource consumption and network coverage requires further investigation. In [16], a drone-assisted cellular networking scheme was proposed to improve coverage performance for machine-type communication services. Cheng et al. in [17] introduced a drone-assisted edge computing architecture for offloading computation-intensive applications. Additionally, a multi-DSC-assisted resource slicing problem for 5G uplink radio access networks was studied in [18] to minimize total resource consumption.

There is a scarcity of literature on resource management in DSC-assisted vehicular networks. Zhang et al. explored software-defined networking (SDN)-based resource management for air-space-ground integrated vehicular networks [19], where local and centralized controllers collaborate to manage resources. He et al. investigated the drone relay problem [20], considering the influence of communication interruption and energy consumption. To support more diversified IoT services in a dynamic network environment, Wu et al. studied a space-air-ground integrated framework for efficient network slicing and content services for vehicular networks [21]. Lyu et al. presented a service-oriented resource slicing framework for space-air-ground integrated vehicular networks to maximize system revenue and stabilize the time-averaged queue [22]. Additionally, Han et al. in [23] developed a drone-aided intelligent transportation system to support low-latency vehicular services. They studied the problem of how to minimize the average peak age of information by optimizing multidrone deployment.

Certain issues require further investigation. For instance, some research assumes that DSCs can provide services to vehicles without the support of GBSs while ignoring the resource consumption that occurs during the interaction between DSCs and GBSs. Moreover, when slicing resources among heterogeneous BSs, it is essential to take into account the traffic features of vehicle services and the distinctive channels used by drones.

### 1.2. Contributions and Organization

In a scenario where multiple DSCs and GBSs coexist, we propose an air-ground integrated spectrum management framework for delay-sensitive applications in 5G and beyond vehicular networks. Our focus is on maximizing network utility under the constraint of delay. This paper makes two main contributions.

- We construct an optimization framework for resource slicing and vehicle association, which takes into account DSC deployment, traffic statistics, inter-DSC interference, and QoS requirements. We formulate a network utility maximization problem using the logarithmic function to determine spectrum slicing ratios and vehicle association patterns. We transform the joint optimization problem into a tractable biconcave maximization problem.
- We develop a convex search algorithm that iteratively solves the transformed problem for vehicle association patterns and spectrum partition with reduced complexity. The algorithm converges to a set of partial optimal solutions. Simulation results demonstrate that the proposed solution outperforms two other resource slicing baseline schemes regarding resource utilization and network throughput.

The follow-up content is arranged in the following sections. Section 2 presents the system model under consideration. Section 3 offers an optimization problem formulation and decomposition. In Section 4, the optimization problem is transformed into a tractable biconcave problem, and an alternate algorithm is proposed to solve the transformed problem. The section also discusses DSC deployment and companion flight policy. The performance evaluation is presented in Section 5. Finally, Section 6 concludes the paper.

Table 1 lists the main notations and variables, and the appendices provide the proof of the propositions and corollaries.

**Table 1.** Main notations and variables.

| Symbols | Definition |
|---|---|
| $a_{i,j,k}$ | Association indicator for vehicle $i$ with the DSC at $v_{j,k}$ |
| $a_{i,m}$ | Association indicator for vehicle $i$ with GBS $m$ |
| $c_{i,j,k}$ | Achievable rates of vehicle $i$ associated with the DSC at $v_{j,k}$ |
| $c_{i,j,k}^{(n)}$ | Achievable rate at vehicle $i$ from the DSC at $v_{j,k}$ for $f_{i,j,k}^{(n)}$ |
| $c_{i,m}$ | Achievable rate at vehicle $i$ from GBS $m$ |
| $c_{i,j,k,m}$ | Achievable rate at the DSC at $v_{j,k}$ from GBS $m$ for vehicle $i$ |
| $c^{(\min)}$ | Minimum rate for a bounded delay violation probability |
| $d_{i,m}$ | Euclidean distance between vehicle $i$ and GBS $m$ |
| $d_{i,j}$ | Horizontal distance between vehicle $i$ and the DSC at $v_{j,k}$ |
| $f_{i,j,k}^{(n)}$ | Amount of spectrum allocated to vehicle $i$ (out of $\alpha_n W$) from the DSC at $v_{j,k}$ |
| $f_{i,m}$ | Amount of spectrum allocated to vehicle $i$ from GBS $m$ |
| $f_{i,j,k,m}$ | Amount of spectrum allocated to vehicle $i$ from GBS $m$ |
| $g_{i,m}$ | Channel gain from GBS $m$ to vehicle $i$ |
| $g_{i,j,k}$ | Channel gain from the DSC at $v_{j,k}$ to vehicle $i$ |
| $g_{j,k,m}$ | Channel gain from GBS $m$ to the DSC at $v_{j,k}$ |
| $\mathcal{V}_m / V_m$ | Set/Num. of candidate DSC positions covered by GBS $m$ |
| $\mathcal{I}_{j,k} / I_{j,k}$ | Set/Num. of vehicles covered by the DSC at $v_{j,k}$ |
| $\mathcal{I}_m / I_m$ | Set/Num. of vehicles covered by GBS $m$ |
| $\mathcal{J}_m / J_m$ | Set/Num. of plane position indexes in the coverage of GBS $m$ |
| $W$ | Available amount of radio spectrum resources to the system |
| $p_m / p_{j,k}$ | Transmit power on GBS $m$/the DSC at $v_{j,k}$ |
| $r_{i,m}$ | Spectrum efficiency at vehicle $i$ from GBS $m$ |
| $r_{i,j,k}^{(n)}$ | Spectrum efficiency at vehicle $i$ from the DSC at $v_{j,k}$ for $f_{i,j,k}^{(n)}$ |
| $r_{j,k,m}$ | Spectrum efficiency at the DSC at $v_{j,k}$ from GBS $m$ |
| $R_k$ | Effective ground coverage radius of the DSC at altitude $z_k$ |
| $v_{j,k}$ | Candidate DSC position $(x_j, y_j, z_k)$ |
| $\alpha_1 / \alpha_2$ | Spectrum slicing ratio for GBS 1/GBS 2 |
| $\alpha_3$ | Spectrum slicing ratio for each DSC |
| $\delta_m$ | Fraction of spectrum resources from $\alpha_m$ for G2V links |
| $\delta_{j,k,m}$ | Fraction of resources from $\alpha_m$ allocated to the DSC at $v_{j,k}$ |
| $\lambda_a$ | Arrival rate of the delay-sensitive packet |
| $\xi_{\text{LoS}}$ | LoS probability threshold for D2V links |
| $\tau_{\text{DU}}$ | Free space path-loss threshold |

## 2. System Model

Consider a two-tier vehicular network with multiple GBSs underlaid by multiple DSCs, as shown in Figure 1. DSCs, as air relays, deployed on demand, can forward GBSs' traffic to target vehicles. When not covered by DSCs, a vehicle chooses to connect to a GBS. Under the coverage of a DSC, a vehicle can choose to connect to the DSC or a GBS. Multiple access types are permitted. Vehicles can access MEC servers via GBSs or DSCs. GBSs can wirelessly charge hovering DSCs [24].

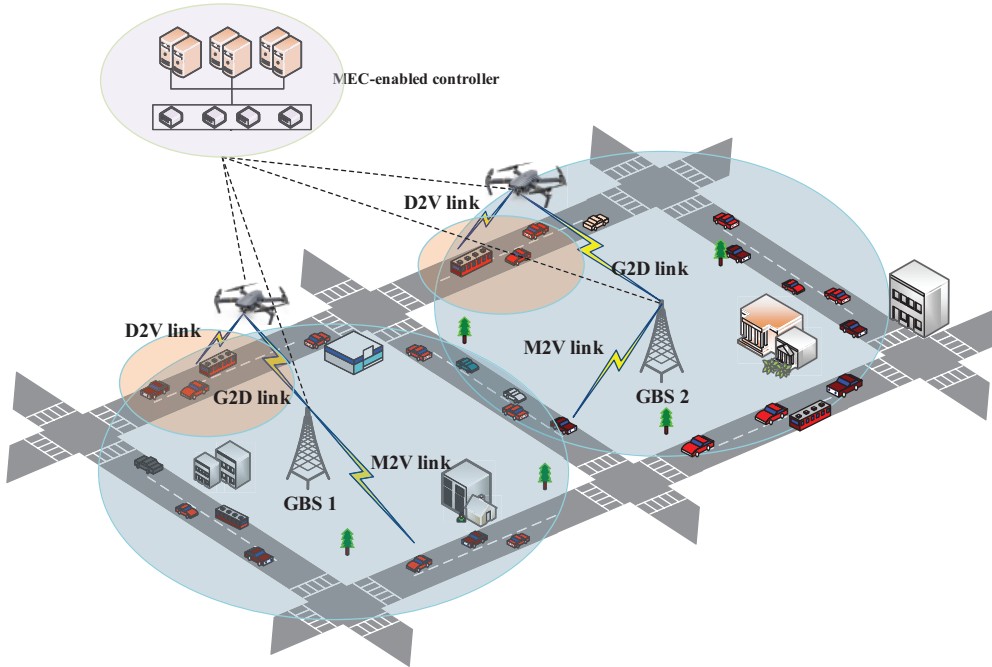

**Figure 1.** Drone-small-cell-assisted cellular vehicular networks.

### 2.1. Resource Slicing Framework

The physical radio resources from GBSs and DSCs are abstracted as a centralized virtual radio resource pool [25,26]. By collecting vehicles' request information, a MEC-enabled controller performs management. GBSs are divided into two groups, denoted by $\mathcal{M}_1$ and $\mathcal{M}_2$, where GBSs in the same group share the same spectrum resources and are not adjacent. Take an example of a two-way lane scenario shown in Figure 2. GBS 1 and GBS 2 are two GBSs from the groups $\mathcal{M}_1$ and $\mathcal{M}_2$, respectively. The system's total available radio spectrum resources are denoted as $W$. Without loss of generality, we consider slicing the spectrum resources among GBS 1, GBS 2, and each DSC. Each DSC reuses spectrum resources to support D2V communications under a distance constraint among DSCs. Then, the spectrum resources are divided into three mutually orthogonal spectrum slices, 1, 2, and 3, with the slicing ratios $\alpha_1$, $\alpha_2$, and $\alpha_3$, and are allocated to GBS 1, GBS 2, and each DSC, satisfying

$$\sum_{n \in \{1,2,3\}} \alpha_n = 1. \tag{1}$$

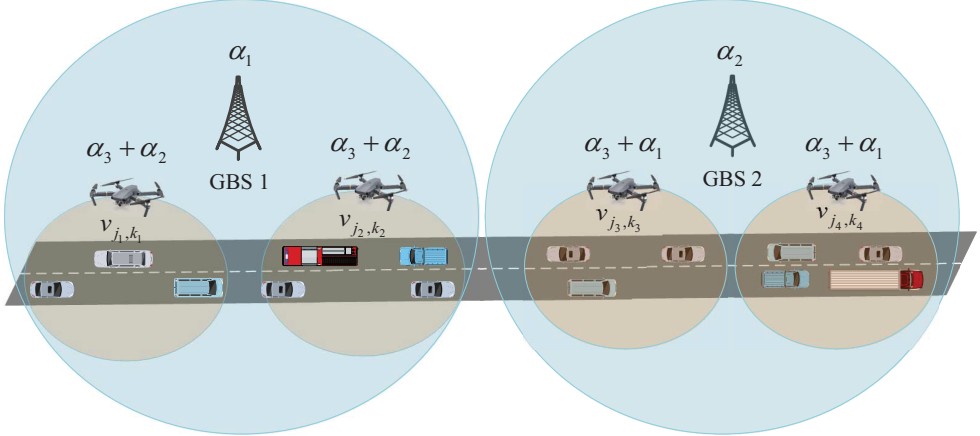

**Figure 2.** Spectrum management framework.

For instance, on the left side of Figure 2, the drone deployed at $v_{j_1,k_1}$ uses spectrum slices 2 and 3, of which slice 3 is shared by each drone, and slice 2 is assigned to GBS 2. Since the drone is far from GBS 2, the interference caused by UAV using slice 2 to GBS 2 is low, with improved resource utilization. Similarly, on the right side of Figure 2, the drone deployed at $v_{j_4,k_4}$ is assigned spectrum slices 1 and 3, where slice 1 comes from GBS 1.

Let $v_{j,k} = (x_j, y_j, z_k)$ denote a drone deployment position. The set of drone deployment positions under the coverage of GBS $m$ is denoted as $\mathcal{V}_m$ with $V_m$ being its cardinality (i.e., the number of available DSCs). The fraction of resources from $\alpha_m$ allocated to associated vehicles to support GBS-to-vehicle (G2V) communications is denoted as $\delta_m$. The fraction of resources from $\alpha_m$ allocated to the DSC associated with GBS $m$ at $v_{j,k} \in \mathcal{V}_m$ for G2D communications is denoted as $\delta_{j,k,m}$ ($m \in \{1,2\}$). The slicing ratios satisfy

$$\alpha_m = \delta_m + \sum_{v_{j,k} \in \mathcal{V}_m} \delta_{j,k,m}. \tag{2}$$

Two-level spectrum reusing is considered. In addition to reusing the resources $W\alpha_3$ among DSCs, we allow the DSCs not covered by a GBS to reuse the GBS's spectrum. The interference to GBSs caused by the DSCs can be controlled via proper deployment of DSCs. Take Figure 2 as an example. The DSCs at $v_{j_1,k_1}$ and $v_{j_2,k_2}$ can reuse the spectrum resource $(\alpha_3 + \alpha_2)W$, and the DSCs at $v_{j_3,k_3}$ and $v_{j_4,k_4}$ can reuse $(\alpha_3 + \alpha_1)W$.

The key to resource slicing is to determine the optimal set of slicing ratios to maximize the entire network utility. After slicing the spectrum resources, the controller allocates the slices to each BS. The resources in each slice is further partitioned among associated vehicles.

## 2.2. Communication Model

As shown in Figure 3, a complex vector space is used to characterize the effect of vehicle direction and speed on distance calculation, where $\vec{d}_{i,m}$ is the distance vector from vehicle $i$ to GBS $m$, and $\vec{v}_i$ represents the velocity vector of vehicle $i$. During a period of length $\Delta t$, the distance vector of vehicle $i$ is expressed as $\vec{v}_i \Delta t$. By the addition or subtraction of complex vectors, the Euclidean distance from vehicle $i$ to GBS $m$ with the vehicle velocity vector $\vec{v}_i$ is defined as

$$d_{i,m} \stackrel{\Delta}{=} \left\| \vec{d}_{i,m} \pm \vec{v}_i \Delta t \right\|. \tag{3}$$

Similarly, in the case where a vehicle is associated with a drone, as shown in Figure 4, the horizontal distance between vehicle $i$ and the drone at the location $v_{j,k}$ is defined as

$$d_{i,j} \stackrel{\Delta}{=} \left\| \vec{d}_{i,j} \pm \vec{v}_i \Delta t \right\|. \tag{4}$$

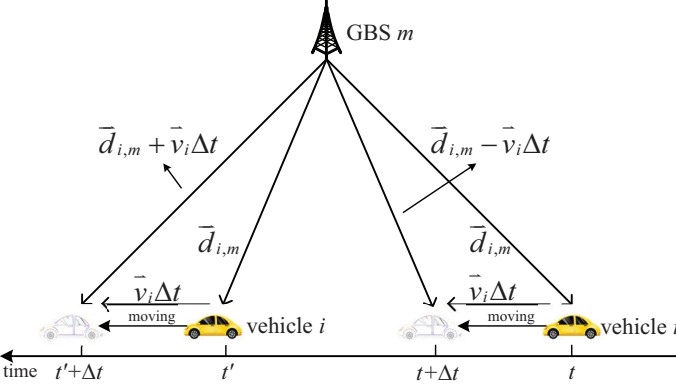

**Figure 3.** Mobility-aware distance calculation when a vehicle is associated with GBS $m$ (Case-1).

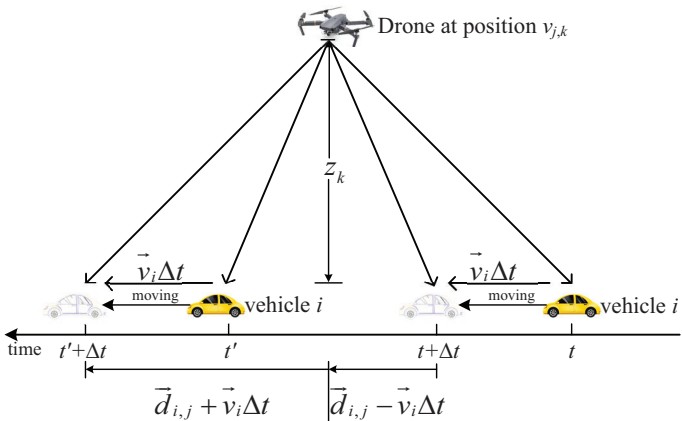

**Figure 4.** Mobility-aware distance calculation when a vehicle is associated with a drone (Case-2).

Let $g_{i,m}$ denote the path loss from GBS $m$ destined for vehicle $i$, which is quantified by substituting (3) into the method described by Ye et al. [27].

With the introduction of LoS probability, we characterize the drone channel. Compared with a non-LoS connection, an LoS connection has less attenuation, which improves spectrum efficiency. By substituting (4) into the aerial channel model proposed in [14,28], we express the LoS probability of the D2V link from a DSC at $v_{j,k}$ to vehicle $i$ as

$$P_{\text{LoS}}(z_j, d_{i,j}) = \frac{1}{1 + e_1 \exp\left(-e_2(\arctan\left(\frac{z_k}{d_{i,j}}\right)) - e_1\right)} \tag{5}$$

where $d_{i,j}$ is the horizontal distance between vehicle $i$ and $v_{j,k}$, and $e_1$ and $e_2$ are constants determined by the environment. Based on [14] and (4), the average path loss of the D2V link forms the DSC at $v_{j,k}$ to vehicle $i$, which is expressed as

$$\begin{aligned} g_{i,j,k} =& 20 \log \sqrt{z_k^2 + d_{i,j}^2} + (\eta_{\text{LoS}} - \eta_{\text{NLoS}}) P_{\text{LoS}}(z_k, d_{i,j}) \\ &+ 20 \log\left(\frac{4\pi\rho}{c}\right) + \eta_{\text{NLoS}}. \end{aligned} \tag{6}$$

In (6), $\eta_{\text{LoS}}$ ($\eta_{\text{NLoS}}$) is the additional loss for LoS (NLoS) links, involving the impacts of shadowing components, $c$ represents the speed of light, and $\rho$ is the carrier frequency.

The vehicle set under the coverage of GBS $m$ is denoted by $\mathcal{I}_m$. Based on the proposed spectrum management framework, vehicle $i \in \mathcal{I}_1$ experiences two kinds of interference: from transmissions of other GBSs in $\mathcal{M}_1$ and of DSCs under the coverage of GBSs in $\mathcal{M}_2$. Let $p_m$ and $p_{j,k}$ represent the transmit power of GBS $m$ and the DSC at $v_{j,k}$. The spectral efficiency at vehicle $i \in \mathcal{I}_1$ from GBS 1 is expressed as

$$r_{i,1} = \log_2\left(1 + \frac{p_1 g_{i,1}}{\sum\limits_{m \in \mathcal{M}_1 \setminus \{1\}} p_m g_{i,m} + \sum\limits_{v_{j,k} \in \mathcal{V}_2} p_{j,k} g_{i,j,k} + \sigma^2}\right) \tag{7}$$

where $\sigma^2$ is the average background noise power. Similarly, the spectrum efficiency at vehicle $i \in \mathcal{I}_2$ from GBS 2, $r_{i,2}$, can be obtained. The achievable transmission rates of vehicle $i$ associated with GBS $m$ can be expressed as

$$c_{i,m} = f_{i,m} r_{i,m} \tag{8}$$

where $f_{i,m}$ is the amount of spectrum (out of $\delta_m W$) allocated to vehicle $i$ from GBS $m$.

For the DSC at $v_{j,k} \in \mathcal{V}_1$, let $f_{i,j,k}^{(2)}$ and $f_{i,j,k}^{(3)}$ be the amount of spectrum allocated to vehicle $i$ out of $\alpha_2 W$ and $\alpha_3 W$. The spectrum efficiency at vehicle $i$ with D2V communications include two parts in terms of $f_{i,j,k}^{(2)}$ and $f_{i,j,k}^{(3)}$, expressed as

$$r_{i,j,k}^{(2)} = \log_2 \left( 1 + \frac{p_{j,k} g_{i,j,k}}{\sum\limits_{m \in \mathcal{M}_1} p_m g_{i,m} + \sum\limits_{v_{j',k'} \in \mathcal{V}_1 \setminus \{v_{j,k}\}} p_{j',k'} g_{i,j',k'} + \sigma^2} \right) \tag{9}$$

and

$$r_{i,j,k}^{(3)} = \log_2 \left( 1 + \frac{p_{j,k} g_{i,j,k}}{\sum\limits_{m \in \{1,2\}} \sum\limits_{v_{j',k'} \in \mathcal{V}_m \setminus \{v_{j,k}\}} p_{j',k'} g_{i,j',k'} + \sigma^2} \right). \tag{10}$$

The achievable transmission rate of vehicle $i$ associated with the DSC at $v_{j,k} \in \mathcal{V}_1$ is the summation of $c_{i,j,k}^{(2)} = f_{i,j,k}^{(2)} r_{i,j,k}^{(2)}$ and $c_{i,j,k}^{(3)} = f_{i,j,k}^{(3)} r_{i,j,k}^{(3)}$. Similarly, denote $f_{i,j,k}^{(1)}$ as the amount of spectrum allocated to vehicle $i$ associated with the DSC at $v_{j,k}$ from $\alpha_1 W$ by the DSC at $v_{j,k}$ under the coverage of GBS 2 ($v_{j,k} \in \mathcal{V}_2$). Then, similar to (9) and (10), the two parts of spectrum efficiencies at the vehicle from the DSC under the coverage of GBS 2, i.e., $r_{i,j,k}^{(1)}$ and $r_{i,j,k}^{(3)}$, can be obtained, and the achievable transmission rate at vehicle $i$ associated with the DSC at $v_{j,k}$ is the summation of $c_{i,j,k}^{(1)} = f_{i,j,k}^{(1)} r_{i,j,k}^{(1)}$ and $c_{i,j,k}^{(3)} = f_{i,j,k}^{(3)} r_{i,j,k}^{(3)}$. If a DSC is associated with GBS $m$, indication variable $b_{j,k,m}$ is set to 1; otherwise 0. Given $b_{j,k,1}$ and $a_{j,k,2}$, the achievable transmission rates of vehicle $i$ associated with the DSC at $v_{j,k}$ can be expressed as

$$c_{i,j,k} = b_{j,k,1} c_{i,j,k}^{(2)} + b_{j,k,2} c_{i,j,k}^{(1)} + b_{j,k,1} c_{i,j,k}^{(3)} + b_{j,k,2} c_{i,j,k}^{(3)}. \tag{11}$$

Let $(x_m, y_m, z_m)$ represent the three-dimensional coordinates of GBS $m$. The distance between $v_{j,k}$ and GBS $m$ is calculated as $d_{j,k,m} = \sqrt{(x_j - x_m)^2 + (y_j - y_m)^2 + (z_k - z_m)^2}$. Since the DSC flying height is usually higher than that of a GBS, the G2D link is an LoS connection. Denote $\gamma$, $\theta_0$, $\eta_0$ as the terrestrial path-loss exponent, angle offset, and excess path-loss offset. Denote $o_1$ and $o_2$ as excess path-loss scalar and angle scalar. The average path loss from GBS $m$ to $v_{j,k}$ is [29]

$$s_{j,k,m} = 10\gamma \log(d_{j,k,m}) + o_1(\theta - \theta_0) \exp\left( \frac{\theta - \theta_0}{o_2} \right) + \eta_0 \tag{12}$$

where $\theta = \arctan(\frac{|z_k - z_m|}{d_{j,k,m}})$ represents the elevation angle between the antennas of the DSC at $v_{j,k}$ and GBS $m$. Similar to (7), the DSC at $v_{j,k}$ associated with GBS 1 experiences two kinds of interference. Then, the spectral efficiency from GBS 1 destined for the DSC at $v_{j,k}$ is expressed as

$$r_{j,k,1} = \log_2 \left( 1 + \frac{p_1 s_{j,k,1}}{\sum\limits_{m \in \mathcal{M}_1 \setminus \{1\}} p_m g_{i,m} + \sum\limits_{v_{j,k} \in \mathcal{V}_2} p_{j,k} g_{i,j,k} + \sigma^2} \right). \tag{13}$$

The spectral efficiency from GBS 2 destined for the DSC at $v_{j,k}$ can be obtained in the same way.

Denote $f_{i,j,k,m}$ as the resources (out of $\delta_{j,k,m} W$) allocated to vehicle $i$ from GBS $m$. When GBS $m$ selects a DSC at $v_{j,k}$ to relay data to vehicle $i$, the achievable transmission rate at the DSC at $v_{j,k}$ can be uniformly expressed as

$$c_{i,j,k,m} = f_{i,j,k,m} r_{j,k,m}. \tag{14}$$

### 2.3. DSC Coverage Model

Consider a realistic drone coverage model. For a DSC placed at $v_{j,k}$, the effective coverage mainly depends on LoS probability and the path-loss threshold in free space [15,28], satisfying

$$\begin{cases} P_{\mathrm{LoS}}(z_k, d_{i,j}) > \xi_{\mathrm{LoS}} \\ \dfrac{4\pi\rho\sqrt{z_k^2 + d_{i,j}^2}}{c} < \tau_{\mathrm{DU}}. \end{cases} \tag{15}$$

In (15), $\xi_{\mathrm{LoS}}$ is the LoS probability threshold for D2V links, and $\tau_{\mathrm{DU}}$ is the free space path-loss threshold, determined by the minimum signal-to-noise ratio for signal decoding.

In the model, flight altitude determines the effective DSC coverage. Similar to the model in [18], the effective ground coverage radius of a DSC flying to a height of $z_k$ can be expressed as

$$R_k = \min\left\{\frac{z_k}{\tan(e_1 - \frac{1}{e_2}\ln\frac{1-\xi_{\mathrm{los}}}{e_1\xi_{\mathrm{los}}})}, \sqrt{\left(\frac{c\tau_{\mathrm{DU}}}{4\pi\rho}\right)^2 - z_k^2}\right\}. \tag{16}$$

Take Figure 5 as an example to explain the influence of the flying height $z_k$ on $R_k$, where $e_1$, $e_2$, $\xi_{\mathrm{LoS}}$, and $\tau_{\mathrm{DU}}$ are set to 4.88, 0.43, and 89 dB and 0.5, respectively. The relationship between height and effective coverage radius is not linear.

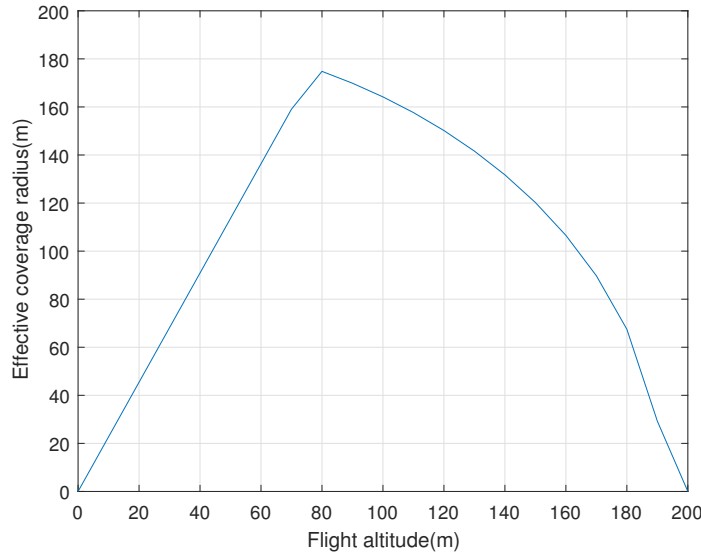

**Figure 5.** Impact of flight altitude on effective coverage radius.

### 2.4. Traffic Model

Consider delay-sensitive traffic (e.g., rear-end collision avoidance, platooning). The average arrival rate and data packet length are denoted as $\lambda_a$ (packet/s) and $L_a$ (bit). The effective bandwidth theory [6,27] is used to calculate the minimum transmission rate to guarantee that the downlink transmission delay exceeding $D^{(\mathrm{max})}$ at most probability $\varepsilon$ is expressed as

$$c^{(\mathrm{min})} = -\frac{L_a \log\varepsilon}{\log(1 - \frac{\log\varepsilon}{\lambda_a D^{(\mathrm{max})}})D^{(\mathrm{max})}}. \tag{17}$$

For downlink transmission to accommodate vehicles' delay-sensitive requests, we can adjust $c^{(\mathrm{min})}$ through resource allocation, providing a probabilistic guarantee for delivery delay and reliability.

### 3. Problem Formulation

In the proposed spectrum management framework, the challenging issue is to determine the optimal spectrum slicing ratios and the association patterns to maximize the aggregate network utility while satisfying the QoS requirement.

Let $\mathcal{I}_{j,k} = \{i \in \mathcal{I} | d_{i,j} \leq R_k\}$ be the set of vehicles located within the effective coverage of the DSC at $v_{j,k}$. If vehicle $i \in \mathcal{I}_{j,k}$ establishes a connection with the DSC at $v_{j,k}$, the indication variable $a_{i,j,k}$ is set to 1; otherwise, 0. If a DSC at $v_{j,k}$ connects to GBS $m$, $b_{j,k,m}$ is set to 1; otherwise, set to 0. Once a DSC flies to GBSs' coverage area, it automatically connects to the GBS with the highest spectral efficiency.

A logarithmic utility function, which is concave and with diminishing marginal utility, is applied to capture proportional fair resource division among heterogeneous BSs. Based on (11), the network utility achieved by all vehicles associated with the DSC at $v_{j,k}$ is expressed as

$$
\begin{aligned}
u_{j,k}&(\mathcal{A}_{j,k}, \mathcal{F}_{j,k}) \\
&= b_{j,k,1} \sum_{i \in \mathcal{I}_{j,k}} a_{i,j,k} \log(c_{i,j,k}^{(2)}) + b_{j,k,2} \sum_{i \in \mathcal{I}_{j,k}} a_{i,j,k} \log(c_{i,j,k}^{(1)}) \\
&+ b_{j,k,1} \sum_{i \in \mathcal{I}_{j,k}} a_{i,j,k} \log(c_{i,j,k}^{(3)}) + b_{j,k,2} \sum_{i \in \mathcal{I}_{j,k}} a_{i,j,k} \log(c_{i,j,k}^{(3)})
\end{aligned}
\tag{18}
$$

where $\mathcal{A}_{j,k} = \{a_{i,j,k} | i \in \mathcal{I}_{j,k}\}$ represents the set of association patterns between vehicles and DSCs and $\mathcal{F}_{j,k} = \{f_{i,j,k}^{(n)} | i \in \mathcal{I}_{j,k}, n \in \{1,2,3\}, a_{i,j,k} = 1\}$ is the strategy set for vehicle-level resource allocation for D2V communications. If vehicle $i$ connects to GBS $m$, the indication variable $a_{i,m}$ is set to 1; otherwise, 0. The network utility achieved by all vehicles associated with GBS $m$ is expressed as

$$
u_m(\mathcal{A}_m, \mathcal{F}_m) = \sum_{i \in \mathcal{I}_m} a_{i,m} \log(c_{i,m}),
\tag{19}
$$

where $\mathcal{A}_m = \{a_{i,m} | i \in \mathcal{I}_m\}$ and $\mathcal{F}_m = \{f_{i,m} | i \in \mathcal{I}_m, a_{i,m} = 1\}$. Given $\mathcal{A}_{j,k}$, the network utility at the DSC at $v_{j,k}$ to relay associated vehicles' traffic is calculated as

$$
u_{j,k,m}(\mathcal{A}_{j,k}, \mathcal{F}_{j,k,m}) = \sum_{i \in \mathcal{I}_{j,k}} a_{i,j,k} \log(c_{i,j,k,m})
\tag{20}
$$

with $\mathcal{F}_{j,k,m} = \{f_{i,j,k,m} | i \in \mathcal{I}_{j,k}, a_{i,j,k} = 1\}$ being the strategy set for vehicle-level resource allocation for the relaying from the DSC at $v_{j,k}$ to GBS $m$.

Based on the logarithmic utility function, an aggregate utility maximization problem is formulated as in $\mathcal{P}1$, under the constraints of DSC deployment, transmission rates, association patterns, and resource partitioning.

$$
\begin{aligned}
\mathcal{P}1 : \underset{\substack{\alpha_1, \alpha_2, \alpha_3, \\ \mathcal{A}_{j,k}, \mathcal{A}_m, \\ \mathcal{F}_{j,k}, \mathcal{F}_m, \mathcal{F}_{j,k,m}}}{\text{Maximize}} \quad & \sum_{v_{j,k} \in \mathcal{V}_1 \cup \mathcal{V}_2} u_{j,k}(\mathcal{A}_{j,k}, \mathcal{F}_{j,k}) \\
+ & \sum_{m \in \{1,2\}} u_m(\mathcal{A}_m, \mathcal{F}_m) + \sum_{m \in \{1,2\}} \sum_{v_{j,k} \in \mathcal{V}_1 \cup \mathcal{V}_2} u_{j,k,m}(\mathcal{A}_{j,k}, \mathcal{F}_{j,k,m})
\end{aligned}
$$

$$a_{i,m}\left(c_{i,m} - c^{(\min)}\right) \geq 0, \forall i \in \mathcal{I}_m, \forall m \in \{1,2\} \tag{21a}$$

$$a_{i,j,k}\left(c_{i,j,k} - c^{(\min)}\right) \geq 0, \forall i \in \mathcal{I}_{j,k}, \forall v_{j,k} \in \mathcal{V}_1 \cup \mathcal{V}_2 \tag{21b}$$

$$a_{i,j,k}\left(c_{i,j,k,m} - c^{(\min)}\right) \geq 0, \forall i \in \mathcal{I}_{j,k}, \forall v_{j,k} \in \mathcal{V}_1 \cup \mathcal{V}_2 \tag{21c}$$

$$\sum_{m \in \{1,2\}} a_{i,m} + \sum_{m \in \{1,2\}} \sum_{v_{j,k} \in \mathcal{V}_m} a_{i,j,k} = 1 \tag{21d}$$

$$\sum_{i \in \mathcal{I}_{j,k}} a_{i,j,k} f_{i,j,k}^{(n)} - \alpha_3 = 0 \tag{21e}$$

$$\sum_{i \in \mathcal{I}_m} a_{i,m} f_{i,m} = \delta_m W, \forall m \in \{1,2\} \tag{21f}$$

s.t.
$$\sum_{i \in \mathcal{I}_{j,k}} a_{i,j,k} f_{i,j,k,m} - \delta_{j,k,m} = 0, \forall v_{j,k} \in \mathcal{V}_1 \cup \mathcal{V}_2 \tag{21g}$$

$$a_{i,j,k}, a_{i,m} \in \{0,1\}, \forall i \in \mathcal{I}_{j,k}, \forall v_{j,k} \in \mathcal{V}_1 \cup \mathcal{V}_2 \tag{21h}$$

$$\sum_{m \in \{1,2\}} \left(\delta_m + \sum_{v_{j,k} \in \mathcal{V}_m} \delta_{j,k,m}\right) + \alpha_3 = 1 \tag{21i}$$

$$\alpha_n, \delta_m, \delta_{j,k,m} \in [0,1], \forall v_{j,k}, n \in \{1,2,3\} \tag{21j}$$

$$f_{i,j,k}^{(n)} \in (0,1), \forall i \in \mathcal{I}_{j,k}, \forall v_{j,k} \in \mathcal{V}_1 \cup \mathcal{V}_2, \forall n \in \{1,2,3\} \tag{21k}$$

$$f_{i,m} \in (0,1), \forall i \in \mathcal{I}_m, \forall m \in \{1,2\} \tag{21l}$$

$$f_{i,j,k,m} \in (0,1), \forall i \in \mathcal{I}_{j,k}, \forall v_{j,k} \in \mathcal{V}_1 \cup \mathcal{V}_2 \tag{21m}$$

The objective function of $\mathcal{P}1$ is the summation of utilities achieved by all vehicles (as receivers) and DSCs (as relays). Constraints (21a)–(21c) ensure that the achievable transmission rate at each receiver is not less than $c^{(\min)}$. Constraint (21d) ensures that each DSC can only connect to one BS. Constraints (21e)–(21g) state the resource allocation requirements for each DSC and GBS. Constraint (21i) is a combination of (1) and (2), reflecting the resource slicing requirement. Constraints (21k)–(21m) demonstrate the requirements on resource allocation for each vehicle.

$\mathcal{P}1$ contains a nonlinear objective function and constraints, a mixed-integer nonlinear programming problem. Each vehicle's spectrum allocation relies on association patterns and resource slicing, making problem solving difficult. For tractability, we first determine the optimal fractions $f_{i,j,k}^{(n)}$ and $f_{i,m}$ and $f_{i,j,k,m}$ allocated to vehicle $i$ from the DSC at $v_{j,k}$ or GBS $m$, given $\alpha_3$, $\delta_m$, and $\delta_{j,k,m}$.

## 4. Solution to $\mathcal{P}1$

In this section, we present a problem approximation method that separates the association schema and resource slice subproblems from Problem $\mathcal{P}1$ to facilitate processing. $\mathcal{P}1$ is transformed into a biconcave optimization problem for mathematical tractability. We then design an alternate concave search algorithm to solve vehicle association patterns iteratively.

### 4.1. Problem Approximation

We simplify $\mathcal{P}1$ by expressing $f_{i,j,k}^{(n)}$, $f_{i,m}$, and $f_{i,j,k,m}$ as a function of $a_{i,j,k}$ to reduce the number of decision variables.

In $\mathcal{P}1$, $u_{j,k}(\mathcal{F}_{j,k})$ is a function of $\mathcal{F}_{j,k}$, indicating the aggregate utility of vehicles associated with the DSC at $v_{j,k}$.

$$
\begin{aligned}
u_{j,k}(\mathcal{F}_{j,k}) = {} & b_{j,k,1} \sum_{i \in \mathcal{I}_{j,k}} a_{i,j,k} \log(f_{i,j,k}^{(2)} r_{i,j,k}^{(2)}) \\
& + b_{j,k,2} \sum_{i \in \mathcal{I}_{j,k}} a_{i,j,k} \log(f_{i,j,k}^{(1)} r_{i,j,k}^{(1)}) \\
& + b_{j,k,1} \sum_{i \in \mathcal{I}_{j,k}} a_{i,j,k} \log(f_{i,j,k}^{(3)} r_{i,j,k}^{(3)}) \\
& + b_{j,k,2} \sum_{i \in \mathcal{I}_{j,k}} a_{i,j,k} \log(f_{i,j,k}^{(3)} r_{i,j,k}^{(3)})
\end{aligned}
\tag{22}
$$

$u_m(\mathcal{F}_m)$ represents the aggregate utility of vehicles associated with GBS $m$, given by

$$
u_m(\mathcal{F}_m) = \sum_{i \in \mathcal{I}_m} a_{i,m} \log(f_{i,m} r_{i,m}).
\tag{23}
$$

$u_{j,k,m}(\mathcal{F}_{j,k,m})$ denotes the utility of relaying vehicles' traffic via the DSC at $v_{j,k}$, given by

$$
u_{j,k,m}(\mathcal{F}_{j,k,m}) = \sum_{i \in \mathcal{I}_{j,k}} a_{i,j,k} \log(f_{i,j,k,m} r_{j,k,m}).
\tag{24}
$$

Based on (22)–(24), $\mathcal{P}1$ can be reformulated as $\mathcal{P}2$.

$$
\begin{aligned}
\mathcal{P}2 : \underset{\mathcal{F}_{j,k}, \mathcal{F}_m, \mathcal{F}_{j,k,m}}{\text{Maximize}} \sum_{v_{j,k} \in \mathcal{V}_1 \cup \mathcal{V}_2} u_{j,k}(\mathcal{F}_{j,k}) + \sum_{m \in \{1,2\}} u_m(\mathcal{F}_m) \\
+ \sum_{m \in \{1,2\}} \sum_{v_{j,k} \in \mathcal{V}_1 \cup \mathcal{V}_2} u_{j,k,m}(\mathcal{F}_{j,k,m})
\end{aligned}
$$

s.t. (21e), (21f), (21g), (21k), (21l), (21m).

Since $\mathcal{F}_{j,k}$, $\mathcal{F}_m$, and $\mathcal{F}_{j,k,m}$ in $\mathcal{P}2$ are thee independent decision variable sets with uncoupled constraints, $\mathcal{P}2$ can be decomposed to three subproblems, $\mathcal{P}2.1$, $\mathcal{P}2.2$, and $\mathcal{P}2.3$:

$$
\mathcal{P}2.1 : \underset{\mathcal{F}_{j,k}}{\text{Maximize}} \sum_{v_{j,k} \in \mathcal{V}_1 \cup \mathcal{V}_2} u_{j,k}(\mathcal{F}_{j,k})
$$

s.t. (21e), (21k).

$$
\mathcal{P}2.2 : \underset{\mathcal{F}_m}{\text{Maximize}} \sum_{m \in \{1,2\}} u_m(\mathcal{F}_m)
$$

s.t. (21f), (21l).

$$
\mathcal{P}2.3 : \underset{\mathcal{F}_{j,k,m}}{\text{Maximize}} \sum_{v_{j,k} \in \mathcal{V}_1 \cup \mathcal{V}_2} u_{j,k,m}(\mathcal{F}_{j,k,m})
$$

s.t. (21g), (21m).

**Proposition 1.** *The solutions for $\mathcal{P}2.1$, $\mathcal{P}2.2$, and $\mathcal{P}2.3$ are (25)–(27).*

$$
f_{i,j,k}^{(n)*} = \frac{a_{i,j,k} \alpha_n W}{\sum_{i' \in \mathcal{I}_{j,k}} a_{i',j,k}} \triangleq f_{j,k}^{(n)*}, n \in \{1, 2, 3\}
\tag{25}
$$

$$
f_{i,m}^{*} = \frac{a_{i,m} \delta_m W}{\sum_{i' \in \mathcal{I}_m} a_{i',m}} \triangleq f_{m}^{*}
\tag{26}
$$

$$f_{i,j,k,m}^* = \frac{a_{i,j,k}\delta_{j,k,m}W}{\sum_{i' \in \mathcal{I}_{j,k}} a_{i',j,k}} \triangleq f_{j,k,m}^* \tag{27}$$

The proof of Proposition 1 is given in Appendix A.1.

Proposition 1 indicates that the optimal fractions of resources allocated to vehicles from the associated GBSs/DSCs are equal partitioning.

From (25)–(27), the values of $f_{i,j,k}^{(n)*}$, $f_{i,m}^*$, and $f_{i,j,k,m}^*$ are determined by $\alpha_m$, $\delta_m$, and $\delta_{j,k,m}$, respectively. Accordingly, we redefine $u_{j,k}(\mathcal{A}_{j,k}, \mathcal{F}_{j,k}^*)$, $u_m(\mathcal{A}_m, \mathcal{F}_m^*)$, and $u_{j,k,m}(\mathcal{A}_{j,k}, \mathcal{F}_{j,k,m}^*)$ as

$$\begin{cases} u_{j,k}(\mathcal{A}_{j,k}, \mathcal{F}_{j,k}^*) \triangleq u_{j,k}(\alpha_3, \mathcal{A}_{j,k}) \\ u_m(\mathcal{A}_m, \mathcal{F}_m^*) \triangleq u_m(\delta_m, \mathcal{A}_m) \\ u_{j,k,m}(\mathcal{A}_{j,k}, \mathcal{F}_{j,k,m}^*) \triangleq u_{j,k,m}(\delta_{j,k,m}, \mathcal{A}_{j,k}). \end{cases} \tag{28}$$

Based on (25)–(28), we reformulate $\mathcal{P}1$ as $\mathcal{P}3$.

$$\mathcal{P}3 : \underset{\substack{\alpha_1,\alpha_2,\alpha_3, \\ \mathcal{A}_{j,k},\mathcal{A}_m}}{\text{Maximize}} \sum_{v_{j,k} \in \mathcal{V}_1 \cup \mathcal{V}_2} u_{j,k}(\alpha_3, \mathcal{A}_{j,k}) + \sum_{m \in \{1,2\}} u_m(\delta_m, \mathcal{A}_m)$$

$$+ \sum_{m \in \{1,2\}} \sum_{v_{j,k} \in \mathcal{V}_1 \cup \mathcal{V}_2} u_{j,k,m}(\delta_{j,k,m}, \mathcal{A}_{j,k})$$

$$\text{s.t.} \begin{cases} a_{i,m}\left(f_m^* r_{i,m} - c^{(\min)}\right), \forall i \in \mathcal{I}_m, \forall m \in \{1,2\} & (29a) \\ a_{i,j,k}(b_{j,k,1}f_{i,j,k}^{(2)*}r_{i,j,k}^{(2)} + b_{j,k,2}f_{i,j,k}^{(1)*}f_{i,j,k}^{(1)} + b_{j,k,1}f_{i,j,k}^{(3)*}r_{i,j,k}^{(3)} & \\ \quad + b_{j,k,2}f_{i,j,k}^{(3)*}r_{i,j,k}^{(3)} - c^{(\min)}) \geq 0, \forall i \in \mathcal{I}_{j,k}, \forall v_{j,k}, \forall n & (29b) \\ a_{i,j,k}\left(f_{j,k,m}^* r_{j,k,m} - c^{(\min)}\right) \geq 0, \forall i \in \mathcal{I}_{j,k}, \forall v_{j,k}, \forall m & (29c) \\ (21\text{h}), (21\text{i}), (21\text{j}) & (29d) \end{cases}$$

As $\{\alpha_3, \mathcal{A}_{j,k}\}$, $\{\delta_m, \mathcal{A}_m\}$, and $\{\delta_{j,k,m}, \mathcal{A}_{j,k}\}$ are coupled under (21i), $\mathcal{P}3$ cannot be decoupled in the same way as $\mathcal{P}2$. $\mathcal{P}3$ is a mixed-integer combinatorial problem, which is difficult to solve. Therefore, it is necessary to transform $\mathcal{P}3$ into a tractable form.

### 4.2. Problem Transformation

To solve $\mathcal{P}2$, we relax 0-1 variables in the sets $\mathcal{A}_{j,k}$ and $\mathcal{A}_m$ to real-valued variables contained in $\widetilde{\mathcal{A}}_{j,k} = \{\widetilde{a}_{i,j,k} | i \in \mathcal{I}_{j,k}\}$ and $\widetilde{\mathcal{A}}_m = \{\widetilde{a}_{i,m} | i \in \mathcal{I}_m\}$, with $\widetilde{a}_{i,j,k} \in [0,1]$ and $\widetilde{a}_{i,m} \in [0,1]$. $\widetilde{a}_{i,m}$ is $a_{i,m}$ with $a_{i,j,k}$ substituted by $\widetilde{a}_{i,j,k}$. $\widetilde{a}_{i,j,k}$ and $\widetilde{a}_{i,m}$ can be considered as the probability of establishing the vehicle association in each spectrum slicing period [27].

**Proposition 2.** *The functions* $u_{j,k}(\alpha_3, \widetilde{\mathcal{A}}_{j,k})$, $u_m(\delta_m, \widetilde{\mathcal{A}}_m)$, *and* $u_{j,k,m}(\delta_{j,k,m}, \widetilde{\mathcal{A}}_{j,k})$ *are biconcave on the decision variable set* $\{\alpha_3, \delta_m, \delta_{j,k,m}\} \times \{\widetilde{\mathcal{A}}_{j,k}, \widetilde{\mathcal{A}}_m\}$.

The proof of Proposition 2 is given in Appendix A.2.

With the variable relaxation, $\mathcal{P}3$ is transformed to $\mathcal{P}4$.

$$\mathcal{P}4 : \underset{\substack{\alpha_1,\alpha_2,\alpha_3, \\ \widetilde{\mathcal{A}}_{j,k},\widetilde{\mathcal{A}}_m}}{\text{Maximize}} \sum_{v_{j,k} \in \mathcal{V}_1 \cup \mathcal{V}_2} u_{j,k}(\alpha_3, \widetilde{\mathcal{A}}_{j,k}) + \sum_{m \in \{1,2\}} u_m(\delta_m, \widetilde{\mathcal{A}}_m)$$

$$+ \sum_{m \in \{1,2\}} \sum_{v_{j,k} \in \mathcal{V}_1 \cup \mathcal{V}_2} u_{j,k,m}(\delta_{j,k,m}, \widetilde{\mathcal{A}}_{j,k})$$

$$\text{s.t.}\begin{cases} \widetilde{a}_{i,m}\left(\widetilde{f}_m^* r_{i,m} - c^{(\min)}\right) \geq 0, \forall i \in \mathcal{I}_m, \forall m \in \{1,2\} & \text{(30a)} \\[2mm] \widetilde{a}_{i,j,k}(b_{j,k,1}\widetilde{f}_{j,k}^{(2)*} r_{i,j,k}^{(2)} + b_{j,k,2}\widetilde{f}_{j,k}^{(1)*} r_{i,j,k}^{(1)} + b_{j,k,1}\widetilde{f}_{j,k}^{(3)*} r_{i,j,k}^{(3)} \\[2mm] \quad + b_{j,k,2}\widetilde{f}_{j,k}^{(3)*} r_{i,j,k}^{(3)} - c^{(\min)}) \geq 0, \forall i \in \mathcal{I}_{j,k}, \forall v_{j,k}, \forall m, \forall n & \text{(30b)} \\[2mm] \widetilde{a}_{i,j,k}\left(\widetilde{f}_{j,k,m}^* r_{j,k,m} - c^{(\min)}\right) \geq 0, \forall i \in \mathcal{I}_{j,k}, \forall v_{j,k}, \forall m & \text{(30c)} \\[2mm] \widetilde{a}_{i,m}, \widetilde{a}_{i,j,k} \in [0,1], \forall i \in \mathcal{I}_{j,k}, \forall v_{j,k}, \forall m & \text{(30d)} \\[2mm] (21\text{i}), (21\text{j}) & \text{(30e)} \end{cases}$$

Constraints (30a) and (30b) belong to linear inequality constraint functions, and constraint (30d) is an affine equality constraint function. Note that $\widetilde{f}_m^*$, $\widetilde{f}_{j,k}^{(n)*}$, and $\widetilde{f}_{j,k,m}^*$ are $f_m^*$, $f_{j,k}^{(n)*}$, and $f_{j,k,m}^*$ with $a_{i,m}$ and $a_{i,j,k}$ substituted by $\widetilde{a}_{i,m}$ and $\widetilde{a}_{i,j,k}$. Constraint (30a) actually indicates that if the DSC at $v_{j,k}$ is associated with GBS $m$ with $a_{i,m} = 1$, the spectrum resource allocation for the vehicle should satisfy

$$a_{i,m} r_{i,m} \geq c^{(\min)} \sum_{i' \in \mathcal{I}_m} \widetilde{a}_{i',m}. \tag{31}$$

Constraints (30b) and (30c) indicate that if $a_{i,j,k} = 1$, the vehicle's resource allocation should satisfy

$$a_{i,j,k} r_{i,j,k}^{(n)} \geq c^{(\min)} \sum_{i' \in \mathcal{I}_{j,k}} \widetilde{a}_{i',j,k} \tag{32}$$

and

$$b_{j,k,1} a_{i,j,k} r_{i,j,k}^{(2)} + b_{j,k,2} a_{i,j,k} r_{i,j,k}^{(1)} + b_{j,k,1} a_{i,j,k} r_{i,j,k}^{(3)} \\ + b_{j,k,2} a_{i,j,k} r_{i,j,k}^{(3)} \geq c^{(\min)} \sum_{i' \in \mathcal{I}_{j,k}} \widetilde{a}_{i',j,k}. \tag{33}$$

Constraints (31)–(33) in $\mathcal{P}4$ indicate the limit on the number of vehicles associated with GBSs/DSCs given $\{\alpha_1, \alpha_2, \alpha_3\}$.

We next simplify $\mathcal{P}4$ to $\mathcal{P}5$ by substituting (30a)–(30c) with (31)–(33), respectively, to make $\mathcal{P}4$ tractable.

$$\mathcal{P}5 : \underset{\substack{\alpha_1, \alpha_2, \alpha_3, \\ \widetilde{\mathcal{A}}_{j,k}, \widetilde{\mathcal{A}}_m}}{\text{Maximize}} \sum_{v_{j,k} \in \mathcal{V}_1 \cup \mathcal{V}_2} u_{j,k}(\alpha_3, \widetilde{\mathcal{A}}_{j,k}) + \sum_{m \in \{1,2\}} u_m(\delta_m, \widetilde{\mathcal{A}}_m)$$
$$+ \sum_{m \in \{1,2\}} \sum_{v_{j,k} \in \mathcal{V}_1 \cup \mathcal{V}_2} u_{j,k,m}(\delta_{j,k,m}, \widetilde{\mathcal{A}}_{j,k})$$
$$\text{s.t. } (31), (32), (33), (21\text{i}), (21\text{j}).$$

Compared with constraint (30a) in $\mathcal{P}4$, constraint (31) in $\mathcal{P}5$ provides the lowest upper bound on the number of vehicles that can be associated with GBS $m$. Similarly, compared with constraints (30b) and (30c) in $\mathcal{P}4$, constraints (32) and (33) in $\mathcal{P}5$ provide the lowest upper bound on the number of vehicles that can be associated with the DSC at $v_{j,k}$.

### 4.3. Algorithm Design

$\mathcal{P}5$ is a biconcave maximization problem due to the biconcave objective function and the set of biconvex constraint functions for the biconvex decision variable set $\{\alpha_3, \delta_m, \delta_{j,k,m}\} \times \{\widetilde{\mathcal{A}}_{j,k}, \widetilde{\mathcal{A}}_m\}$. We first summarize the concavity property of $\mathcal{P}5$.

**Corollary 1.** *The objective function of $\mathcal{P}4$ is a biconcave function on the variable set $\{\alpha_3, \delta_m, \delta_{j,k,m}\} \times \{\widetilde{\mathcal{A}}_{j,k}, \widetilde{\mathcal{A}}_m\}$, and $\mathcal{P}4$ is a biconcave optimization problem.*

The proof of Corollary 1 is given in Appendix A.3.

**Corollary 2.** *Algorithm 1 can converge to a set of optimal solutions $\{\alpha_3^*, \delta_m^*, \delta_{j,k,m}^*\} \times \{\widetilde{\mathcal{A}}_{j,k}^*, \widetilde{\mathcal{A}}_m^*\}$.*

The proof of Corollary 2 is given in Appendix A.4.

---

**Algorithm 1:** Alternate_search_algorithm

---

**Input** : $\vartheta$; candidate set for $\{\alpha_1, \alpha_2, \alpha_3\}$.
**Output:** Optimal spectrum slicing ratios $\{\alpha_1^*, \alpha_2^*, \alpha_3^*\}$ with $\Theta_m^* = \{\delta_m^*, \delta_{j_1,k_1,m}^*,$
$\quad\quad \delta_{j_2,k_2,m}^*, \ldots, \}$ split from $\alpha_m^*$ ($m \in \{1, 2\}$); optimal association pattern set
$\quad\quad \widetilde{\mathcal{A}}^*$.

1   $t \leftarrow 0; u^{(t)} \leftarrow 0; u^{(t+1)} \leftarrow 0;$
2   **while** $||u^{(t+1)} - u^{(t)}|| \geq \vartheta$ **do**
3      Initialize candidate values for $\Theta_m$ and $\Theta_{j,k,m}$ given $\alpha_m^{(t)}$ ($m \in \{1, 2\}$) and $\widetilde{\mathcal{A}}^{(t)}$;
4      $\Theta_1^{(t)}$ and $\Theta_2^{(t)} \leftarrow$ solving $\mathcal{P}3$ given $\widetilde{\mathcal{A}}^{(t)}$ and $\{\alpha_1^{(t)}, \alpha_2^{(t)}, \alpha_3^{(t)}\}$.
5      $\widetilde{\mathcal{A}}^\dagger \leftarrow$ solving $\mathcal{P}3$ given $\alpha_3^{(t)}, \Theta_1^{(t)}$, and $\Theta_2^{(t)}$;
6      Obtain $\mathcal{F}^\dagger$ given $\widetilde{\mathcal{A}}^\dagger, \alpha_3^{(t)}, \Theta_1^{(t)}$, and $\Theta_2^{(t)}$;
7      **if** *no solutions for $\mathcal{P}3$* **then**
8          Reinitialize until no solutions found; break;
9      **else**
10          $\widetilde{\mathcal{A}}^{(t+1)} \leftarrow \widetilde{\mathcal{A}}^\dagger;$
11          $\alpha_3^\dagger, \Theta_1^\dagger$, and $\Theta_2^\dagger \leftarrow$ solving $\mathcal{P}3$ given $\widetilde{\mathcal{A}}^{(t+1)}$;
12          Obtain $\mathcal{F}^\dagger$ given $\widetilde{\mathcal{A}}^{(t+1)}, \alpha_3^\dagger, \Theta_1^\dagger$, and $\Theta_2^\dagger$;
13          **if** *no solutions for $\mathcal{P}3$* **then**
14              Reinitialize until no solutions found; break;
15          **else**
16              $\alpha_3^{(t+1)} \leftarrow \alpha_3^\dagger;$
17              $\Theta_1^{(t+1)}$ and $\Theta_2^{(t+1)} \leftarrow \Theta_1^\dagger$ and $\Theta_2^\dagger;$
18              $\mathcal{F}^{(t+1)} \leftarrow \mathcal{F}^\dagger;$
19              Obtain $u^{(t+1)}$ with $\alpha_3^{(t+1)}, \Theta_1^{(t+1)}, \Theta_2^{(t+1)}, \widetilde{\mathcal{A}}^{(t+1)}$, and $\mathcal{F}^{(t+1)}$ at the $t$th iteration;
20          **end**
21          $t \leftarrow t + 1;$
22      **end**
23 **end**

---

By exploring the biconcavity, we develop an alternate search algorithm to solve $\mathcal{P}5$, summarized in Algorithm 1. The main logic is to iteratively solve optimal association patterns $\{\widetilde{\mathcal{A}}_{j,k}^*, \widetilde{\mathcal{A}}_m^*\}$ and optimal spectrum slicing ratios $\{\alpha_3^*, \delta_m^*, \delta_{j,k,m}^*\}$ to maximize the objective function. In the $(t+1)$th iteration, given a spectrum slicing ratio set, $\{\alpha_3^{(t)}, \delta_m^{(t)}, \delta_{j,k,m}^{(t)}\}$, and an association pattern set, $\{\widetilde{\mathcal{A}}_{j,k}^{(t)}, \widetilde{\mathcal{A}}_m^{(t)}\}$, from the $t$th iteration, $\mathcal{P}5$ is solved to find a better association pattern set, $\{\widetilde{\mathcal{A}}_{j,k}^\dagger, \widetilde{\mathcal{A}}_m^\dagger\}$ with $\{\mathcal{F}_{j,k}, \mathcal{F}_m, \mathcal{F}_{j,k,m}\}$. To control computational complexity, we reduce the space of candidate slicing ratios. Let $u^{(t)}$ denote the maximum objective function value with $\{\widetilde{\mathcal{A}}_{j,k}^{(t)}, \widetilde{\mathcal{A}}_m^{(t)}\}$ at the beginning of the $t$th iteration. If the difference between $u^{(t+1)}$ and $u^{(t)}$ is less than the threshold $\vartheta$, the iteration stops, and the algorithm converges to a set of optimal solutions, $\{\alpha_3^*, \delta_m^*, \delta_{j,k,m}^*\}$ and $\{\widetilde{\mathcal{A}}_{j,k}^*, \widetilde{\mathcal{A}}_m^*\}$; otherwise, start the next iteration until it converges. As stated in Corollary 2, the algorithm can converge.

## 5. Performance Evaluation

Extensive simulations are carried out to verify the effectiveness of the proposed solution. All the simulations are carried out using MATLAB and Python and run on a computer with an Intel Core i3 processor and 8 GB RAM. Consider a scenario with two adjacent GBSs and multiple DSCs. Each GBS's height and coverage radius are set to 10 and 800 m. The DSC flying height range on each x-y plane coordinate is [0, 200 m] with an adjacent height interval of 10 m, and the horizontal movement range on the x-y coordinate plane is set to [−1600 m,1600 m]. The DSC's effective coverage at different heights is determined by (16). The number of DSCs and DSCs' flight altitude determine the drone coverage ratio. Each GBS (DSC) has the same downlink transmit power of 46 dBm (24 dBm). Each lane's vehicle density range is set to [0.05, 0.5] v/m, where the minimum vehicle distance is 5 m. The average rate $\lambda_a$ of packet arrivals is 4 packet/s. The packet length ($L_a$) is 1048 bit. The packet deadline bound $D^{(\max)}$ and deadline bound violation probability $\varepsilon$ are 0.001 s and $10^{-3}$. Table 2 lists other important parameters.

**Table 2.** Parameter settings.

| Parameters | Values |
|---|---|
| GBS altitude $m$ ($z_m$) | 10 m |
| Coverage radius of each GBS ($R_m$) | 800 m |
| Transmit power of GBS $m$ ($p_m$) | 46 dBm |
| Transmit power of the DSC at $v_{j,k}$ ($p_{j,k}$) | 24 dBm |
| Urban environment parameter ($e_1/e_2$) | 4.88/0.43 |
| Excess path-loss scalar/angle scalar($o_1/o_2$) | −23.29/4.14 |
| Additional loss for LoS/NLoS links ($\eta_{\text{LoS}}/\eta_{\text{NLoS}}$) | 0.1/21 |
| Terrestrial path-loss exponent ($\gamma$) | 3.04 |
| Angle offset ($\theta_0$) | 3.61 |
| Excess path-loss offset ($\eta_0$) | 20.7 |
| Carrier frequency ($f$) | 3.5 GHz |
| LoS probability threshold for D2V links ($\xi_{\text{LoS}}$) | 0.5 |
| Free space path-loss threshold ($\tau_{\text{DU}}$) | 89 dB |
| Packet arrival rate ($\lambda_a$) | 4 pkt/s |
| Packet length ($L_a$) | 1048 bit |
| Packet delay bound ($D^{(\max)}$) | 0.001 s |
| Delay bound violation probability ($\varepsilon$) | $10^{-3}$ |
| Stop criterion ($\vartheta$) | 0.01 |

The proposed scheme is categorized as versions I and II. The former is a full-featured version with flight altitude adaptation as in [18], while the latter does not allow DSCs to reuse GBSs' spectrum resources.

For comparison, we provide two baseline schemes:

- **Maximization-SINR (max-SINR)** [14], in which the DSC deployment with flight altitude adaptation aims to maximize the aggregate spectrum efficiency;
- **Maximization-DSC-coverage (max-Cov)** [15], in which each DSC always maintains the height that maximizes the effective coverage.

Each baseline is further categorized as versions I and II. The former uses the same dynamic DSC deployment as the proposed scheme, while the latter is with static deployment.

### 5.1. Impact of Available Spectrum Resources

The first simulation examines network throughput, presented as the system's aggregate transmission rate. The average vehicle density is set to 0.1 vehicles/meter (v/m). Figure 6 compares the throughputs achieved by different approaches where two DSCs are deployed. As more resources are allocated, the amount provisioned per vehicle increases, leading to higher transmission rates. The throughput of the proposed scheme rises more rapidly than other schemes.

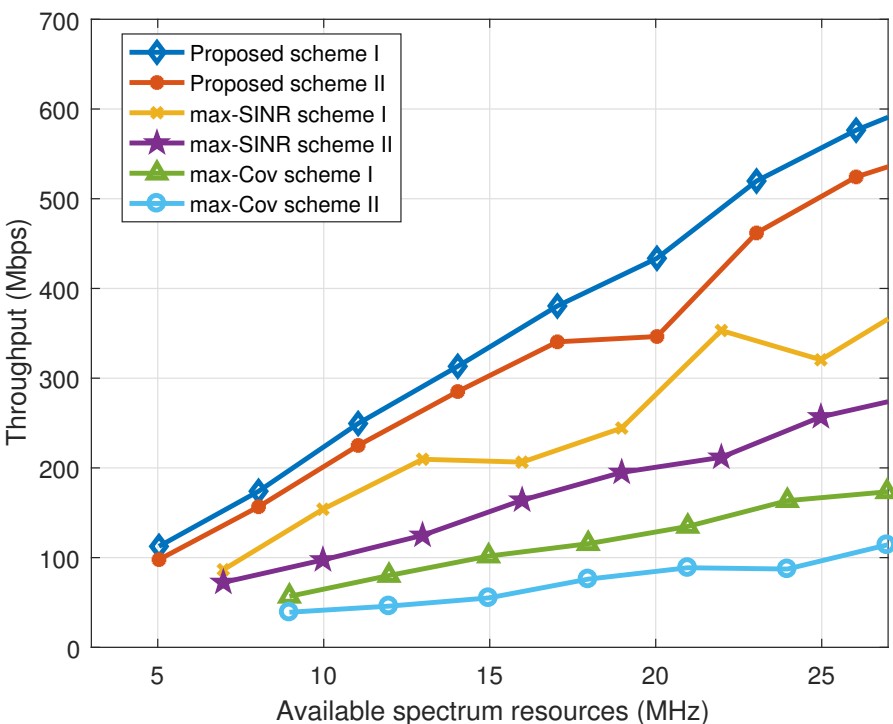

**Figure 6.** Impact of spectrum resources on different schemes.

Additionally, dynamic DSC deployment outperforms static deployment. Specifically, the proposed scheme's minimum spectrum resource requirement is 5 MHz, while at least 7 and 8 MHz are needed by the max-SINR and max-Cov schemes, respectively. Owing to efficient spectrum reuse and slicing, the proposed scheme's network throughput is on average over 30% higher than the max-SINR scheme and over 45% higher on average than the max-Cov scheme.

In Figure 7, the starting point on the left represents the lower bound of resources required by different strategies under QoS constraints. As more DSCs are added, more vehicles can connect to DSCs, and overall spectrum utilization increases. Resource partitioning depends largely on DSC deployment and vehicle distribution. The proposed scheme achieves higher throughput than baselines given the same resource budget. The results demonstrate the proposed scheme's ability to improve network throughput through dynamic resource allocation. The gains are achieved by maximizing spectrum reuse and slicing efficiency under QoS requirements.

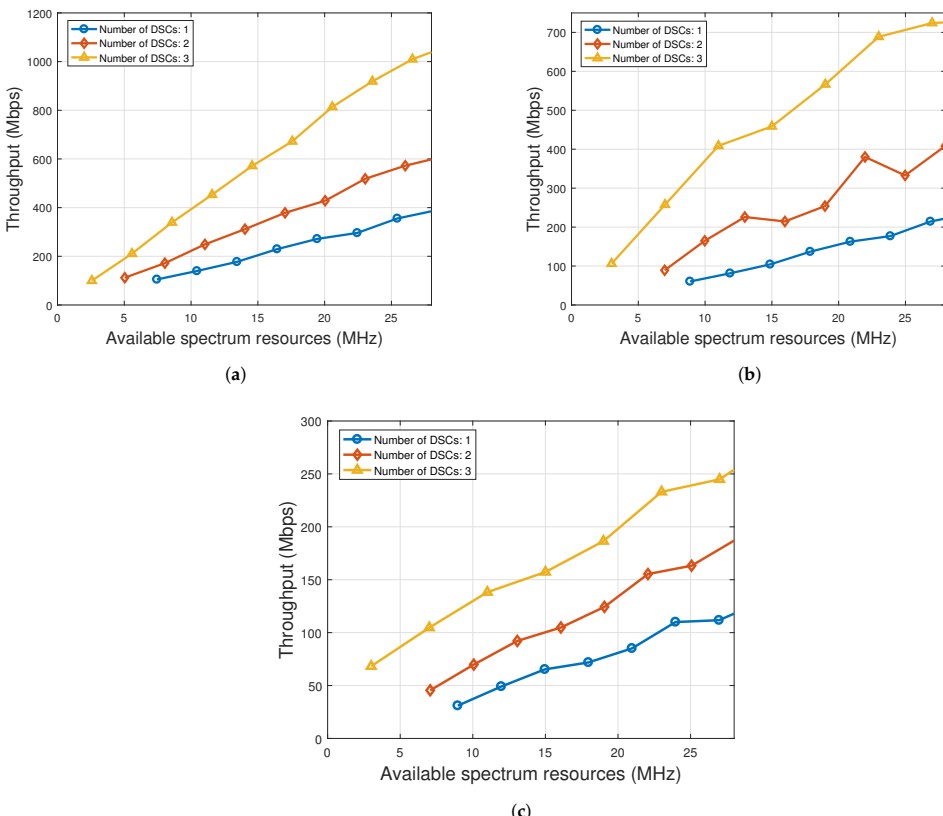

**Figure 7.** Impact of spectrum resources and the number of DSCs: (**a**) proposed scheme I, (**b**) max-SINR scheme I, (**c**) max-Cov scheme I.

### 5.2. Impact of Vehicle Density

In the following simulation, the amount of available spectrum resources is 20 MHz. Figure 8 shows the impact of average vehicle density on the minimum spectrum resource consumption for different methods. Increasing vehicle density leads to greater demand for spectrum resources. The proposed scheme's minimum spectrum consumption is on average over 15% lower than the max-SINR scheme and over 25% lower on average than the max-Cov scheme, with a slower growth trend as vehicle density rises.

In Figure 9, increasing the number of DSCs can significantly improve spectrum utilization and throughput. The proposed method can more effectively leverage DSCs for spectrum reuse and partitioning. From Figure 10, the resource slicing ratios are adjusted accordingly as the average vehicle density grows from 0.05 to 0.5 v/m. A higher vehicle density makes spectrum resources more scarce, prompting more vehicles to connect to DSCs and increasing the resource portion allocated to DSCs. The proposed intelligent resource management is efficient, especially in dense vehicular scenarios. The dynamic spectrum slicing balances the resource allocation between GBSs and DSCs based on real-time demand.

The cooperation of drones enables the network to accommodate more vehicle access. Nonetheless, connected vehicles are a dynamic environment. The management of UAV deployment, resource allocation, and vehicle association must be brought into a unified framework to play the role of different platforms.

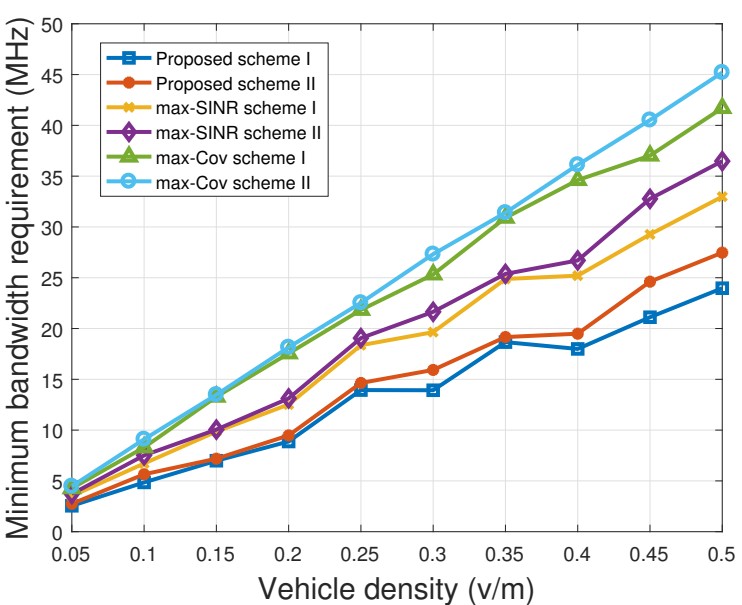

**Figure 8.** Impact of vehicle density on different schemes.

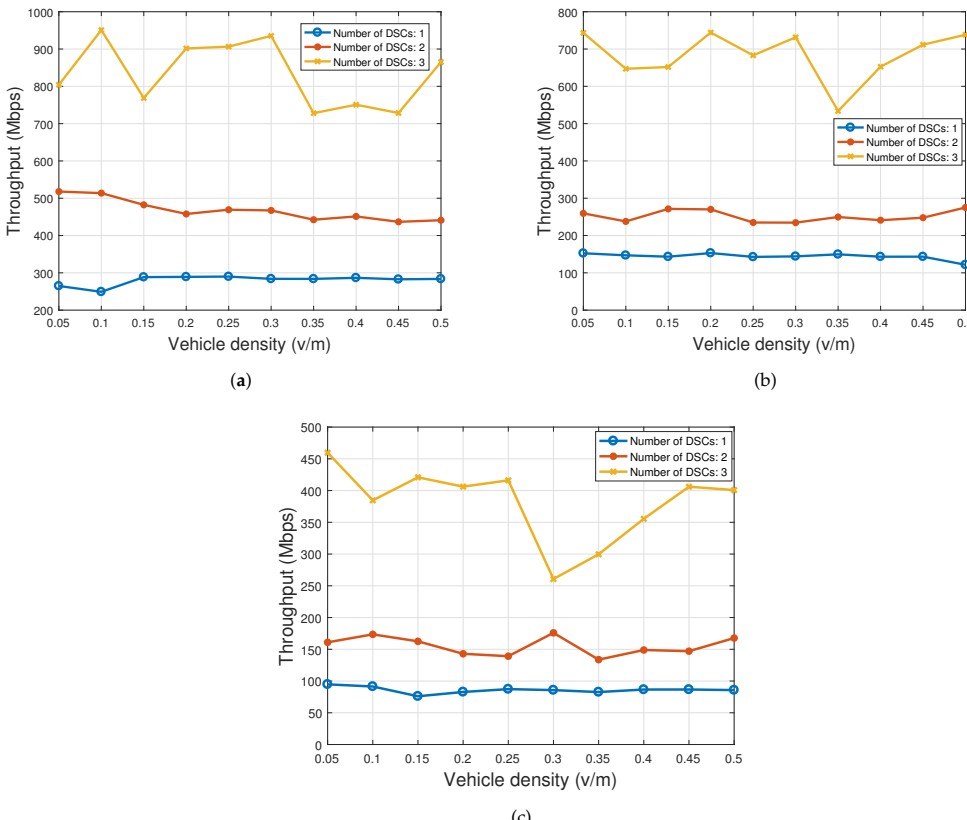

**Figure 9.** Impact of vehicle density and the number of DSCs on throughput: (**a**) proposed scheme I, (**b**) max-SNR scheme I, (**c**) max-Cov scheme I.

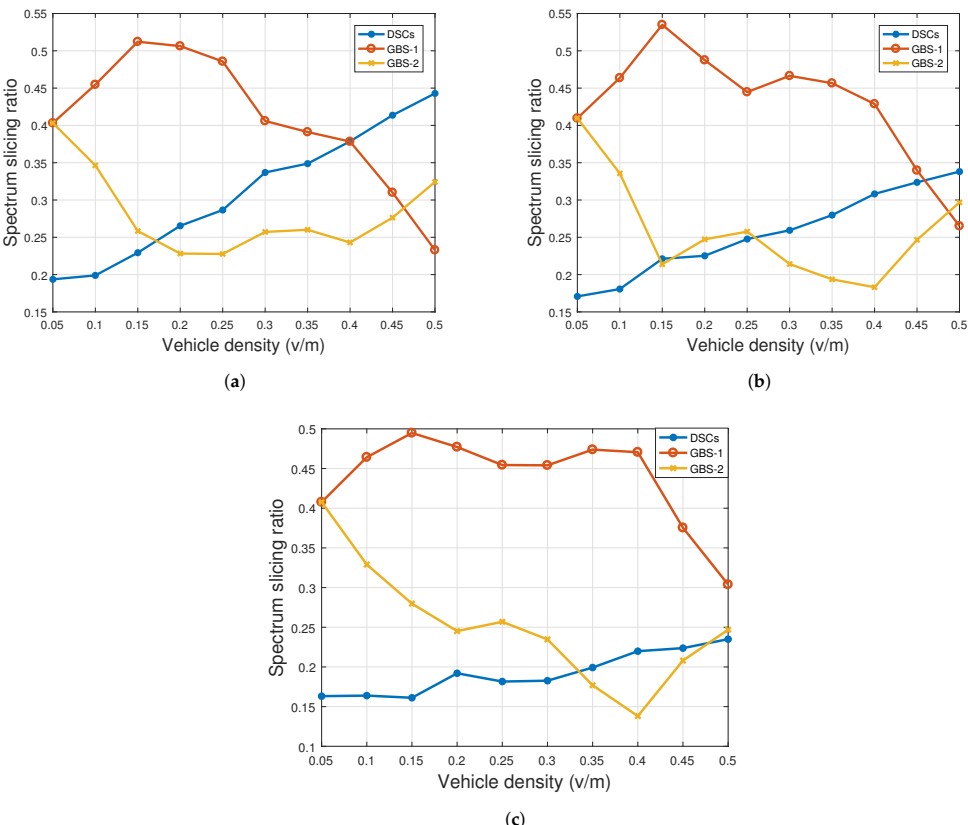

**Figure 10.** Impact of vehicle density on spectrum slicing ratios: (**a**) proposed scheme I, (**b**) max-SNR scheme I, (**c**) max-Cov scheme I.

## 6. Conclusions

In this paper, we have proposed a spectrum management framework for drone-assisted vehicular networks. The goal is to maximize network utility subject to QoS constraints. The network utility maximization problem is formulated to determine vehicle–DSC association patterns and spectrum partitioning among heterogeneous BSs. The optimization problem is further transformed into a tractable biconcave form, followed by an alternate search algorithm to obtain optimal spectrum slicing ratios and association patterns. Simulation results demonstrate that the proposed method has advantages in throughput and spectrum utilization. The proposed framework is scalable and has the potential to be used to support content distribution in air-ground integrated vehicular networks. Our ongoing work will design a distributed machine-learning-based resource slicing method to adapt to large-scale vehicular network scenarios where multiple services coexist.

**Author Contributions:** H.S. put forward the original ideas and performed the research, Y.H. performed the experiments and analyzed the data, T.W. provided useful comments, and G.B. raised the research question and reviewed this paper. All authors have read and agreed to the published version of the manuscript.

**Funding:** This research was funded by the National Natural Science Foundation of China under Grants 61502230 and 61501224, the Natural Science Foundation of Jiangsu Province under Grant BK20201357, and the Six Talent Peaks Project in Jiangsu Province under Grant RJFW-020.

**Data Availability Statement:** Not applicable.

**Conflicts of Interest:** The authors declare no conflicts of interest.

## Appendix A.

*Appendix A.1. Proof of Proposition 1*

Since DSCs reuse the slice $W_2$, and each vehicle can only connect to one BS, $\mathcal{P}2.1$ can be decoupled into $(V_1 + V_2)$ items, each for one DSC. According to (22), each item has four parts. For the first part, we construct the subproblem $\mathcal{P}3.1.1$.

$$\mathcal{P}3.1.1 : \text{Maximize} \; b_{j,k,1} \sum_{i \in \mathcal{I}_{j,k}} a_{i,j,k} \log(f_{i,j,k}^{(2)} r_{i,j,k}^{(2)})$$

$$\text{s.t.} \begin{cases} \sum_{i \in \mathcal{I}_{j,k}} a_{i,j,k} f_{i,j,k}^{(2)} - \alpha_2 = 0 & \text{(A1a)} \\ f_{i,j,k}^{(2)} \in (0,1), \forall i \in \mathcal{I}_{j,k} & \text{(A1b)} \end{cases}$$

The objective function of $\mathcal{P}3.1.1$ can be further derived as

$$\begin{aligned} & b_{j,k,1} \sum_{i \in \mathcal{I}_{j,k}} a_{i,j,k} \log(f_{i,j,k}^{(2)} r_{i,j,k}^{(2)}) \\ & = b_{j,k,1} \log \prod_{i \in \mathcal{I}_{j,k}} \left( r_{i,j,k}^{(2)} \right)^{a_{i,j,k}} + b_{j,k,1} \log \prod_{i \in \mathcal{I}_{j,k}} \left( f_{i,j,k}^{(2)} \right)^{a_{i,j,k}} \end{aligned} \tag{A2}$$

In (A2), $r_{i,j,k}^{(2)}$ can be seen as a constant independent of $f_{i,j,k}^{(2)}$. Therefore, $\mathcal{P}3.1.1$ is equivalent to

$$\mathcal{P}3.1.2 : \underset{\mathcal{F}_{j,k}}{\text{Maximize}} \prod_{i \in \mathcal{I}_{j,k}} \left( f_{i,j,k}^{(2)} \right)^{a_{i,j,k}}, \text{ s.t. (A1a), (A1b)}.$$

Since the geometric average is not larger than the arithmetic average, we have

$$\sqrt[\sum_{i \in \mathcal{I}_{j,k}} a_{i,j,k}]{\prod_{i \in \mathcal{I}_{j,k}} \left( f_{i,j,k}^{(2)} \right)^{a_{i,j,k}}} \le \frac{\sum_{i \in \mathcal{I}_{j,k}} \left( f_{i,j,k}^{(2)} \right)^{a_{i,j,k}}}{\sum_{i \in \mathcal{I}_{j,k}} a_{i,j,k}}. \tag{A3}$$

Under (A1a) and (A1b), the optimal resource allocation for vehicle $i$ associated with the DSC at $v_{j,k}$ is obtained by (27).

The remaining proofs for (26) and (27) are similar, which are omitted here.

*Appendix A.2. Proof of Proposition 2*

Substituting (25) into $b_{j,k,1} \sum_{i \in \mathcal{I}_{j,k}} \widetilde{a}_{i,j,k} \log(f_{j,k}^{(2)*} r_{i,j,k}^{(2)})$, we have

$$\begin{aligned} & b_{j,k,1} \sum_{i \in \mathcal{I}_{j,k}} \widetilde{a}_{i,j,k} \log(f_{j,k}^{(2)*} r_{i,j,k}^{(2)}) = b_{j,k,1} \sum_{i \in \mathcal{I}_{j,k}} \widetilde{a}_{i,j,k} \log(W \alpha_n r_{i,j,k}^{(n)}) \\ & - b_{j,k,1} \sum_{i \in \mathcal{I}_{j,k}} \widetilde{a}_{i,j,k} \log \left( \sum_{i' \in \mathcal{I}_{j,k}} \widetilde{a}_{i',j,k} \right). \end{aligned} \tag{A4}$$

By stating an equation for the coefficients using the indices $i_l$ and $i_{l'}$, we express the Hessian matrix of (A4) regarding $\widetilde{\mathcal{A}}_{j,k}$ as

$$\mathbf{H}_{i_l, i_{l'}} = \frac{\partial^2 (b_{j,k,1} \sum_{i \in \mathcal{I}_{j,k}} \widetilde{a}_{i,j,k} \log(f_{j,k}^{(2)*} r_{i,j,k}^{(2)}))}{\partial \widetilde{a}_{i_l,j,k} \partial \widetilde{a}_{i_{l'},j,k}} = - \frac{q_{j,k}}{\sum_{i \in \mathcal{I}_{j,k}} \widetilde{a}_{i,j,k}}. \tag{A5}$$

For any nonzero vector $\mathbf{s} = [s_1, s_2, \ldots, s_{I_{j,k}}] \in \mathbb{R}^{I_{j,k}}$, in the case of $q^*_{j,k} = 1$, we have

$$\mathbf{s}^T \mathbf{H}_{i_l, i_{l'}} \mathbf{s} = -\frac{q_{j,k} \sum_{i \in \mathcal{I}_{j,k}} s_i^2}{\sum_{i \in \mathcal{I}_{j,k}} \widetilde{a}_{i,j,k}} \leq 0. \tag{A6}$$

Since the Hessian matrix is negative definite, $b_{j,k,1} \sum_{i \in \mathcal{I}_{j,k}} \widetilde{a}_{i,j,k} \log(f^{(2)*}_{j,k} r^{(2)}_{i,j,k})$ is a concave function with respect to $\widetilde{\mathcal{A}}_{j,k}$ for any given $\alpha_3$, and the reverse is also true.

Substituting (27) into $b_{j,k,1} \sum_{i \in \mathcal{I}_{j,k}} \widetilde{a}_{i,j,k} \log(f^{(2)*}_{j,k} r^{(2)}_{i,j,k})$, we have

$$\begin{aligned}
b_{j,k,1} &\sum_{i \in \mathcal{I}_{j,k}} \widetilde{a}_{i,j,k} \log(f^{(2)*}_{j,k} r^{(2)}_{i,j,k}) \\
&= b_{j,k,m} \sum_{i \in \mathcal{I}_{j,k}} \widetilde{a}_{i,j,k} \log\left( W \delta_{j,k,m} r_{j,k,m} \right) \\
&\quad - b_{j,k,m} \sum_{i \in \mathcal{I}_{j,k}} \widetilde{a}_{i,j,k} \log\left( \sum_{v_{j,k} \in \mathcal{V}_m} b_{j,k,m} \sum_{i' \in \mathcal{I}_{j,k}} \widetilde{a}_{i',j,k} \right)
\end{aligned} \tag{A7}$$

The element in the Hessian matrix of $u_{j,k,m}(\delta_{j,k,m}, \widetilde{A}_{i,j,k})$ with respect to $\widetilde{A}_{i,j,k}$ is expressed as

$$\mathbf{H}_{i_l, i_{l'}} = \frac{\partial^2 u_{j,k,m}(\delta_{j,k,m}, \mathcal{A}_{j,k})}{\partial \widetilde{a}_{i_l,j,k} \partial \widetilde{a}_{i_{l'},j,k}} = -\frac{a_{j,k,m}}{\sum_{v_{j,k} \in \mathcal{V}_m} a_{j,k,m} \sum_{i \in \mathcal{I}_{j,k}} \widetilde{a}_{i,j,k}}. \tag{A8}$$

In the case of $v_{j,k} \in \mathcal{V}_m$ and $a_{j,k,m} = 1$, we have

$$\mathbf{s}^T \mathbf{H}_{i_l, i_{l'}} \mathbf{s} = -\frac{\sum_{i \in \mathcal{I}_{j,k}} s_i^2}{\sum_{v_{j,k} \in \mathcal{V}_m} \sum_{i \in \mathcal{I}_{j,k}} \widetilde{a}_{i,j,k}}. \tag{A9}$$

Since the matrix is negative definite, $b_{j,k,1} \sum_{i \in \mathcal{I}_{j,k}} \widetilde{a}_{i,j,k} \log(f^{(2)*}_{j,k} r^{(2)}_{i,j,k})$ is a concave function in terms of $\widetilde{\mathcal{A}}_{j,k}$ for any given $\delta_{j,k,m}$, and the reverse is also true.

The proof for $u_m(\widetilde{\mathcal{A}}_m)$ is similar, which is omitted here.

*Appendix A.3. Proof of Corollary 1*

The objective function of $\mathcal{P}5$ is a non-negative linear combination of a set of biconcave functions, which also belongs to a biconcave function on the variable set $\{\alpha_3, \delta_m, \delta_{j,k,m}\} \times \{\widetilde{\mathcal{A}}_{j,k}, \widetilde{\mathcal{A}}_m\}$ [30].

*Appendix A.4. Proof of Corollary 2*

$\{\alpha_3, \delta_m, \delta_{j,k,m}\} \times \{\widetilde{\mathcal{A}}_{j,k}, \widetilde{\mathcal{A}}_m\}$ are closed sets, and the objective function of $\mathcal{P}4$ is continuous on its domain. To verify the uniqueness of $\mathcal{F}^{(t+1)}$ and $\widetilde{\mathcal{A}}^{(t+1)}$ at the end of the $t$-th iteration, we refer to the proof of Corollary 1 that, given $\{\alpha_3, \delta_m, \delta_{j,k,m}\}$, the objective function of $\mathcal{P}4$ is a concave function of $\{\widetilde{\mathcal{A}}_{j,k}, \widetilde{\mathcal{A}}_m\}$. Conversely, given $\widetilde{\mathcal{A}}$, the objective function is also concave in terms of $\Theta$. Therefore, Algorithm 1 can converge to $\{\alpha^*_3, \delta^*_m, \delta^*_{j,k,m}\}$ and $\{\widetilde{\mathcal{A}}^*_{j,k}, \widetilde{\mathcal{A}}^*_m\}$.

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
