# Peer review of "Joint Resource Slicing and Vehicle Association for Drone-Assisted Vehicular Networks†"

_drones, doi:10.3390/drones7080534_

Round 1
Reviewer 1 Report
This paper proposes a joint resource slicing and vehicle association method for drone-assisted vehicular networks. The considered problem is transformed and then solved by biconcave optimization. Simulations are performed to showcase the potential benefits of the proposed method. This paper is overall well-organized with easy-to-follow English. I have the following comments for the authors, just for reference.
1) General concept comments.
1. Nomenclature. It is a good practice to collect the variables used in this paper in Table 1. Given that there are also many abbreviations in this paper, the authors are suggested to put the abbreviations as well in the nomenclature. Please also check the format requirement of the journal, i.e., where to locate the nomenclature, as some of the MDPI journals requires it at the end of the paper. Moreover, caption and table lines are rarely seen for nomenclature. Please consider reformulate it.
2. The paper is very informative. For some readers, it will be welcome to provide more explanations. For instance, based on Fig. 2, it is still hard to understand the description of “Then, the spectrum resources are divided into three mutually orthogonal spectrum slices, with slicing ratios α1, α2, and α3, and are allocated to GBS 1, GBS 2, and each DSC …”. What elements are orthogonal to each other? It is not straightforward to understand.
3. Figs. 3 and 4 need improving. They have the same caption, which is confusing for the readers. Also, based on Fig. 2, the GBS seems to be located in the same horizontal plane with the vehicles, whereas the drones are in a space. In that case, the distance calculation should be different as two- and three-dimensional Euclidean distance are required.
4. The authors are suggested to give more explanations of key concepts, for instance, the line-of-sight and the constraints in the considered scenario. In specific, although related references are cited, how is Eq. (5) formulated? What is LoS if it is denoted in Figs. 3 and 4?
5. Optimization algorithm lacks basic settings. The only listed parameter of tolerance is not enough. Please enrich the settings where possible. Also, please briefly state how the problems are solved, via some toolboxes or by some other codes?
6. Optimization results lacks detailed analysis. In the introduction, it is declared that the biconcave optimization is intended to reduce the computational complexity compared to the original parameter optimization problem. To support this statement, one should list the required computational source of the proposed method, and, if possible, the same quantity of the original problem as a benchmark.
2) Specific comments referring to line numbers, tables, or figures
1. Figs. 7, 9, and 10: the legends and label are too small for them. Please consider use the blank margin on the left-hand side. In the Latex template of the journal, this operation is supported.
2. Algorithm 1, Line 16: should the “end” here be removed?
3. Appendix: it is usually numbered separately by A.1, A.2… instead of 7.1 , 7.2, …
4. Mind the size of the braces and brackets. For instance, the braces in Eq. (16) is too small to accommodate the elements therein.
Author Response
Responses to Reviewer 1’s Comments
Journal Title: Drones
Paper Title: Joint Resource Slicing and Vehicle Association for Drone-Assisted Vehicular Networks
We would like to thank the editor and all the reviewers for the valuable comments and suggestions on our manuscript. The manuscript has been carefully revised by taking into consideration of all the comments and suggestions from the reviewers. Our responses are provided in detail in the following, and are incorporated in the revised manuscript.
RESPONSE TO REVIEWER 1’s COMMENTS
Comment 1: Nomenclature. It is a good practice to collect the variables used in this paper in Table 1. Given that there are also many abbreviations in this paper, the authors are suggested to put the abbreviations as well in the nomenclature. Please also check the format requirement of the journal, i.e., where to locate the nomenclature, as some of the MDPI journals requires it at the end of the paper. Moreover, caption and table lines are rarely seen for nomenclature. Please consider reformulate it.
Response: Thank you for your suggestion. I carefully checked the abbreviations in the full text to ensure that all abbreviations have relevant notes. In addition, I checked several papers in the Drones Journal, and found no nomenclature.
Comment 2: The paper is very informative. For some readers, it will be welcome to provide more explanations. For instance, based on Fig. 2, it is still hard to understand the description of “Then, the spectrum resources are divided into three mutually orthogonal spectrum slices, with slicing ratios α1, α2, and α3, and are allocated to GBS 1, GBS 2, and each DSC …”. What elements are orthogonal to each other? It is not straightforward to understand.
Response: Thanks for your advice. We have added relevant explanations below Eq. (1).
Comment 3: Figs. 3 and 4 need improving. They have the same caption, which is confusing for the readers. Also, based on Fig. 2, the GBS seems to be located in the same horizontal plane with the vehicles, whereas the drones are in a space. In that case, the distance calculation should be different as two- and three-dimensional Euclidean distance are required.
Response: Thank you very much for your suggestion. We have rearranged the names of Figure 3 and Figure 4. We consider the instantaneous velocity vector of the vehicle under the road network. In a road network environment, the vehicle's driving direction will not change significantly in a short period. Therefore, we estimate the vehicle movement distance in a two-dimensional coordinate system.
Comment 4: The authors are suggested to give more explanations of key concepts, for instance, the line-of-sight and the constraints in the considered scenario. In specific, although related references are cited, how is Eq. (5) formulated? What is LoS if it is denoted in Figs. 3 and 4?
Response: We have cited references 15 and 29 in the text, which contain the measurement and calculation process for Eq. (5). Figs. 3 and 4 are used to understand how to estimate the distance traveled by the vehicle, independent of LOS.
Comment 5: Optimization algorithm lacks basic settings. The only listed parameter of tolerance is not enough. Please enrich the settings where possible. Also, please briefly state how the problems are solved, via some toolboxes or by some other codes?
Response: We used Matlab and Python to simulate and run the algorithm. We have added relevant clarification in the Performance Evaluation Section.
2) Specific comments referring to line numbers, tables, or figures
Comment 1: Figs. 7, 9, and 10: the legends and label are too small for them. Please consider use the blank margin on the left-hand side. In the Latex template of the journal, this operation is supported.
Response: We have reformatted Figs. 7, 9, and 10.
Comment 2: Algorithm 1, Line 16: should the “end” here be removed?
Response: Thank you for reminding me. We have changed the description framework of a latex algorithm and have removed "end" in Algorithm 1.
Comment 3: Appendix: it is usually numbered separately by A.1, A.2… instead of 7.1 , 7.2, …
Response: We have adjusted to the form of A.1, A.2.
Comment 4: Mind the size of the braces and brackets. For instance, the braces in Eq. (16) is too small to accommodate the elements therein.
Response: Thanks. We have modified the brace form of Eq. (16) as requested.

Reviewer 2 Report
There are no comments or suggestions for the authors
Author Response
Thank you for your recognition of our research work.
Reviewer 3 Report
The following are suggestions towards improvements in the paper:
1. The methodology section could be strengthened in the form of work flow diagram detailing the steps in the implementation.
2. Some observations and inferences can be made based on the results presented.
3. This could lead to more concrete conclusions
4. There could improvement in the logical flow of the paper.
Author Response
Responses to Reviewer 3’s Comments
Journal Title: Drones
Paper Title: Joint Resource Slicing and Vehicle Association for Drone-Assisted Vehicular Networks
We would like to thank the editor and all the reviewers for the valuable comments and suggestions on our manuscript. The manuscript has been carefully revised by taking into consideration of all the comments and suggestions from the reviewers. Our responses are provided in details in the following, and are incorporated in the revised manuscript.
RESPONSE TO REVIEWER 1’s COMMENTS
Comment 1: The methodology section could be strengthened in the form of work flow diagram detailing the steps in the implementation.
Response: To reduce complexity, the optimization problem is transformed to a tractable biconcave maximization problem. Then, an alternative search algorithm is designed to solve the transformed problem for a set of optimal slicing ratios and optimal vehicle association patterns. We describe the above ideas and implementation step by step in Section 4.
Comment 2: Some observations and inferences can be made based on the results presented.
Response: We have added relevant content as requested at the end of Subsection 5.3.
Comment 3: This could lead to more concrete conclusions
Response: We have supplemented the Conclusion Section by exploring the scalability and generalizability of the proposed framework.
Comment 4: There could improvement in the logical flow of the paper.
Response: We have supplemented the general description of the solution between Section 4 and Subsection 4.1 to understand the subsequent concrete implementation.

Reviewer 4 Report
This paper presents a joint resource slicing and vehicle association framework for a drone-small-cell-assisted air-ground integrated network. It uses mixed-integer nonlinear programming to maximize network utility while considering traffic statistics, quality-of-service (QoS) constraints, vehicle locations, and inter-drone interference. The proposed scheme outperforms two baseline schemes in terms of throughput and spectrum utilization.
Overall this paper demonstrated solid result and merit to the field, so I suggest publish in its current form.
Author Response
Thank you for your high recognition of our research work.
Round 2
Reviewer 1 Report
The authors have made efforts to improve the paper quality. The reviewer thinks that the paper may be accepted if the following minor comments are suitably addressed.
1. In line 376 of PP. 16, only the information of the used software is provided. Please also supplement the specific version of MATLAB, the used toolbox (if any), and the hardware information, for instance, CPU, RAM, et al.
2. Please also use (A1), (A2) … for the equations in the appendix.
Author Response
RESPONSE TO REVIEWER 1’s COMMENTS
Comment 1: In line 376 of PP. 16, only the information of the used software is provided. Please also supplement the specific version of MATLAB, the used toolbox (if any), and the hardware information, for instance, CPU, RAM, et al.
Response: Thank you for your suggestion. We have added relevant content about computer configurations.
Comment 2: Please also use (A1), (A2) … for the equations in the appendix.
Response: We have improved the numbering of appendices by your suggestion.